# Reciprocal interactions between tumour cell populations enhance growth and reduce radiation sensitivity in prostate cancer

Marcin Paczkowski [1], Warren W. Kretzschmar [2,3,10], Bostjan Markelc[4], Stanley K. Liu [5], Leoni A. Kunz-Schughart [6,7], Adrian L. Harris [4,8,12], Mike Partridge[4,12], Helen M. Byrne [1,12✉] & Pavitra Kannan [4,9,11,12✉]

Intratumoural heterogeneity (ITH) contributes to local recurrence following radiotherapy in prostate cancer. Recent studies also show that ecological interactions between heterogeneous tumour cell populations can lead to resistance in chemotherapy. Here, we evaluated whether interactions between heterogenous populations could impact growth and response to radiotherapy in prostate cancer. Using mixed 3D cultures of parental and radioresistant populations from two prostate cancer cell lines and a predator-prey mathematical model to investigate various types of ecological interactions, we show that reciprocal interactions between heterogeneous populations enhance overall growth and reduce radiation sensitivity. The type of interaction influences the time of regrowth after radiation, and, at the population level, alters the survival and cell cycle of each population without eliminating either one. These interactions can arise from oxygen constraints and from cellular cross-talk that alter the tumour microenvironment. These findings suggest that ecological-type interactions are important in radiation response and could be targeted to reduce local recurrence.

[1] Mathematical Institute, University of Oxford, Oxford, UK. [2] School of Engineering Sciences in Chemistry Biotechnology and Health, Department of Gene Technology, Science for Life Laboratory, KTH Royal Institute of Technology, Stockholm, Sweden. [3] Center for Hematology and Regenerative Medicine (HERM), Department of Medicine Huddinge, Karolinska Institutet, Stockholm, Sweden. [4] CRUK and MRC Oxford Institute for Radiation Oncology, University of Oxford, Oxford, UK. [5] Sunnybrook Research Institute and Departments of Medical Biophysics and Radiation Oncology, University of Toronto, Toronto, ON, Canada. [6] OncoRay—National Center for Radiation Research in Oncology, Faculty of Medicine and University Hospital Carl Gustav Carus, TU Dresden and Helmholtz-Zentrum, Dresden, Rossendorf, Germany. [7] National Center for Tumor Diseases (NCT), Partner Site, Dresden, Germany. [8] Weatherall Institute of Molecular Medicine, University of Oxford, Oxford, UK. [9] Department of Microbiology, Tumor and Cell Biology, Karolinska Institutet, Stockholm, Sweden. [10]Present address: Center for Hematology and Regenerative Medicine (HERM), Department of Medicine Huddinge, Karolinska Institutet, Stockholm, Sweden. [11]Present address: Department of Microbiology, Tumor and Cell Biology, Karolinska Institutet, Stockholm, Sweden. [12]These authors jointly supervised this work: Adrian L. Harris, Mike Partridge, Helen M. Byrne, Pavitra Kannan. ✉email: helen.byrne@maths.ox.ac.uk; pk@sciencesylt.com

ocal recurrence remains a challenge in the management of localised prostate cancer with external beam radiotherapy[1]. Intratumoral heterogeneity (ITH) is one factor that likely contributes to recurrence. This heterogeneity stems in part from genetic variations: primary lesions contain multiple genetically and clonally distinct foci that are often maintained in recurrent and metastatic prostate tumours[2–5]. Phenotypic variations, such as changes in metabolism, also affect therapeutic response among prostate tumour clones[1,6–9]. Given this genotypic and phenotypic diversity, a population-based approach in targeting ITH with radiotherapy may help to reduce local recurrence.

One recent strategy to target ITH has been to model and treat tumours as complex ecosystems in which treatment-sensitive and resistant cell populations interact with each other[10,11]. Similar to interactions between species in other ecological systems[12], those between tumour cell populations can be competitive, antagonistic (i.e., predatorial), or mutualistic (i.e., cooperative), depending on the microenvironmental pressures or communication between cell populations[11]. The interactions may alter the survival of each tumour cell population and, consequently, the entire tumour's response to treatment[13–17]. By mapping the interactions and survival of sensitive and resistant populations under different treatment regimens, alternative treatment regimens that reduce local recurrence can be identified[15,18].

Although well-studied with regard to chemotherapy, little is known about whether ecological interactions could contribute to prostate cancer recurrence to radiotherapy. Conventional approaches to modelling radiotherapy outcomes typically neglect ITH in two important ways. Firstly, the linear-quadratic model used to predict tumour control probability contains single, fixed values for the radiation sensitivity parameters (i.e., $\alpha$ and $\beta$, the constants for cell killing in the linear and quadratic phases, respectively)[19]. Simulations of tumour response to radiotherapy have shown, however, that modulating the radiation dose to account for populations with different radiation sensitivities reduces recurrence in silico[20]. Secondly, cell kill events are modelled as a Poisson process, with subsequent events occurring independently of one another[19,21]. Although the discovery of ecological-type interactions in tumours suggests that cell kill events may not occur independently, simulations and experimental evidence demonstrating the impact of these interactions on radiation response are lacking.

Characterising ecological-type interactions between tumour populations with differing radiation sensitivities may lead to new approaches that account for ITH at the radiotherapy treatment planning stage. Here, we evaluate whether such interactions occur between tumour cell populations with different radiation sensitivities and whether these interactions affect tumour growth kinetics and radiation response. Since prostate tumours develop resistance through multiple genetic mechanisms and often contain multiple clones[22], we employed polyclonal tumour populations from two prostate cell lines to better simulate clinical scenarios of radio-recurrence. Given the complexity of ecological interactions, we developed an interdisciplinary approach to: (1) quantify the changes in tumour growth (before and after radiation) that result from ecological interactions between tumour cell populations, (2) determine the type of interaction (e.g., competition vs. cooperation), and (3) identify whether microenvironmental pressures and/or cell–cell communication mediate these interactions. Our results show that different types of ecological interactions enhance the growth of untreated tumours before and after radiation and that these interactions may arise in part from microenvironmental pressures and cellular cross-talk.

## Results

**Characterisation of polyclonal tumour populations with different radiation sensitivities.** Parental (i.e., sensitive) and radiation-resistant populations from two prostate cell lines, PC3 and DU145, were used to evaluate the impact of ecological-type interactions on radiation response. The polyclonal radio-resistant populations from both cells lines were previously characterised to have increased clonogenic survival, increased invasive potential, and reduced $G_2M$ arrest after radiation through different genetic mechanisms[23,24]. Here, after verifying that unlabelled radioresistant populations were morphologically different (Supplementary Fig. 1A) and had significantly higher 2D clonogenic survival (Supplementary Fig. 1B), we labelled the cell populations with fluorescent proteins to track the dynamics of each population under different conditions in our study and checked whether phenotypic differences were maintained. In both fluorescently labelled cell lines, the 2D clonogenic survival of the radioresistant populations was ≥40% higher after 2 Gy (PC3, $P = 0.003$; DU145, $P < 0.001$) than that of the parental population (Fig. 1A). Proliferation also increased by ≥27% after 2 Gy (PC3, $P_{adj} = 0.026$, day 3; DU145, $P_{adj} = 0.003$, day 3) and after 6 Gy radiation (PC3, $P_{adj} = 0.005$, day 9; DU145, $P_{adj} = 0.022$, day 2; Fig. 1B). Radioresistant populations were also more resistant to death induced by the DNA damaging agent cisplatin (PC3, parental $IC_{50} = 6.5$ [95% CI: 4.9–7.9], radioresistant $IC_{50} = 13.0$ [95% CI: 10.9–15.1], $P < 0.0001$; DU145, parental $IC_{50} = 1.8$ [95% CI: 1.6–2.0], radioresistant $IC_{50} = 6.6$ [95% CI: 5.9–7.3], $P < 0.0001$; Fig. 1C). Together, these data confirm that the polyclonal, labelled radioresistant populations have intrinsically lower radiation sensitivity than their parental counterparts, making them suitable to represent the recurring disease in our study. The term 'radioresistant' is henceforth used to describe the cell populations with intrinsically lower radiation sensitivity than their parental counterparts.

**Mixed cultures of tumour populations have enhanced growth and altered survival in the absence of radiation.** To assess whether mixed cultures had altered growth characteristics, we measured the growth of unirradiated 3D tumour spheroids comprised of homogeneous or mixed cultures of parental and radioresistant populations (seeded 1:1). This in vitro model was selected because it mimics important features of solid tumours[25] such as hypoxia, which renders tumours 2–3 times more resistant to radiation. Compared to homogeneous parental spheroids, volumes of mixed PC3 and DU145 spheroids increased by >30% ($P_{adj} < 0.001$, day 10) and by >22% ($P_{adj} < 0.001$, day 15, Fig. 2A), respectively. Since the change in growth could result from the radioresistant population outgrowing the parental one, we quantified the proportions of each population isolated from mixed spheroids over time using flow cytometry (Supplementary Fig. 2). On day 15 of growth, radioresistant cells isolated from mixed PC3 spheroids comprised 60.9 ± 9.6% of total cells, while those isolated from mixed DU145 spheroids comprised 22.7 ± 12.2% of total cells (Fig. 2B).

Analysis of each population's survival revealed that different dynamics were at play in the two cell lines. In mixed PC3 spheroids, cell death was reduced by ≥14% (parental, $P_{adj} = 0.038$; RR, $P_{adj} = 0.045$) for both populations on day 15 (Fig. 2C). However, in mixed DU145 spheroids, cell death was enhanced by 26% for the radioresistant population on day 5 ($P_{adj} = 0.01$, Fig. 2C), which could have led to an early survival advantage for the parental population. Thus, despite the contrasting population dynamics between the two cell lines, these results indicate that overall tumour growth, and survival of

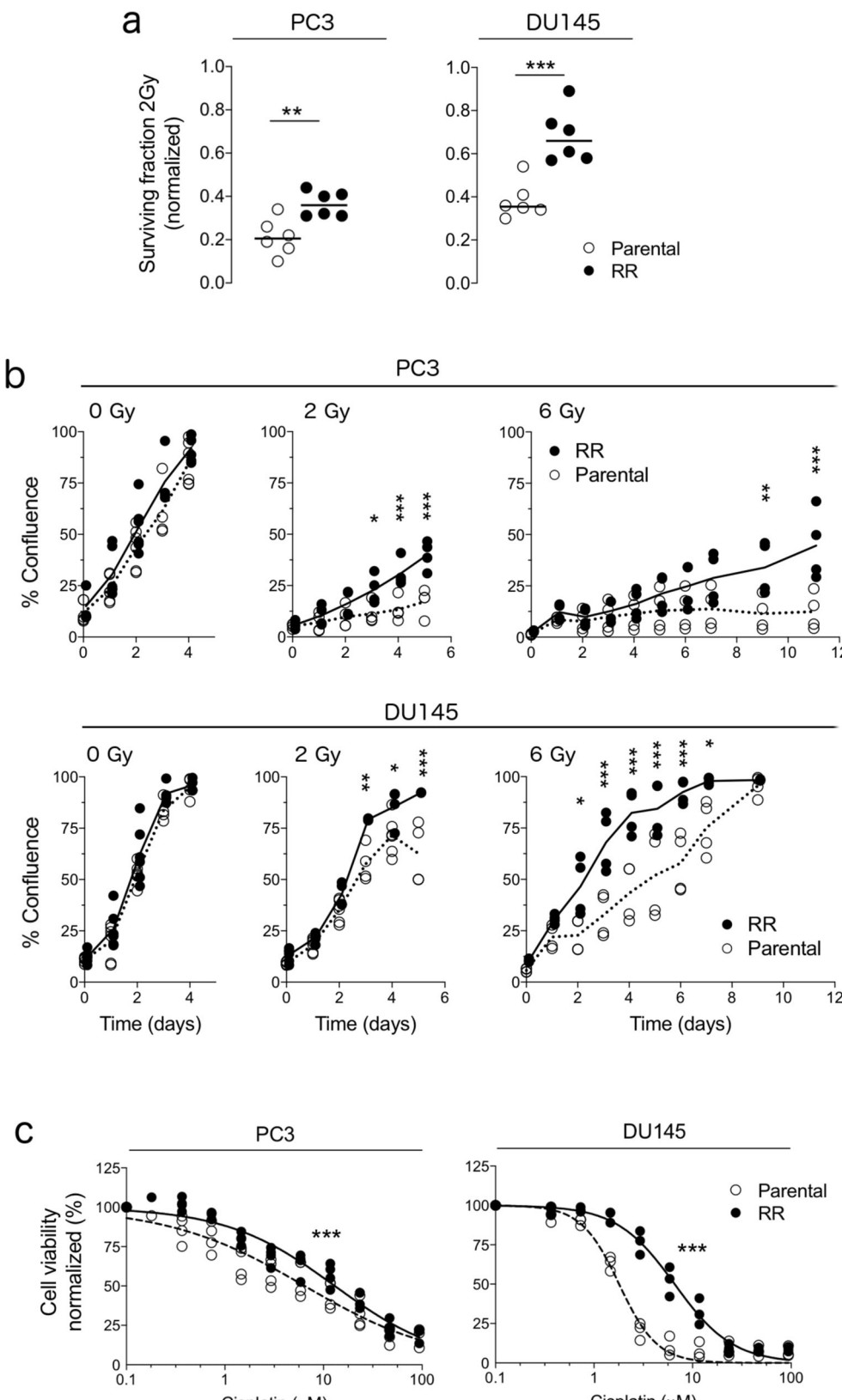

parental populations, in particular, are enhanced in unirradiated tumours with mixed populations.

**Increased growth of unirradiated mixed tumours results from ecological-type interactions.** We used mathematical modelling to determine whether enhanced growth of mixed spheroids could have resulted from changes in proliferation capacity or from cellular interactions between the two populations. Baseline characteristics of each population were calculated by fitting the logistic growth model[26] to data from homogeneous spheroids to estimate the growth rate $r$, the carrying capacity $K$, and the initial

**Fig. 1 Radiation resistant cell populations from two prostate cell lines have lower radiation sensitivity than parental counterparts. a** Surviving fraction at 2 Gy, as determined by clonogenic assay, for fluorescently labelled parental (white circles) and radioresistant (RR, black circles) populations from PC3 and DU145 cell lines. Surviving fraction values were normalised to 0 Gy. Bar indicates the median; data points represent biological replicates ($n = 6$). $**P <$ 0.01, $***P < 0.001$, as determined by unpaired, one-tailed, $t$ test. **b** Growth curves of parental and RR populations grown as monolayers without and with radiation. Bar indicates the mean; data points represent biological replicates ($n = 6$ for 0 Gy; $n = 4$ for 2 and 6 Gy). $*P_{adj} < 0.05$, $**P_{adj} < 0.01$, $***P_{adj} <$ 0.001, as determined by a mixed-effects model with Sidak correction. **c** Percent cell viability (normalised to untreated wells) of parental and RR populations grown as monolayers in response to increasing concentrations of cisplatin (PC3, $n = 4$ experiments; DU145, $n = 3$ experiments). The line indicates the best fit, concentration-response model (normalised response with variable slope). $***P < 0.001$ for PC3 and DU145, as determined by extra sum-of-squares $F$-test.

volume $V_0$ (Fig. 3A). In PC3 spheroids, the radioresistant population had higher values of $r$ and $K$ than the parental population, while in DU145 spheroids, it had lower $r$ and higher $K$ values; values of $V_0$ were not different between cell lines (Table 1). The growth rate and carrying capacity were held fixed at these values in subsequent in silico experiments.

To characterise the growth dynamics and type of ecological interactions between the populations in mixed spheroids, we fit the Lotka–Volterra (or predator–prey) model to data from the mixed spheroids (Fig. 3A). In this model, the interaction parameters, $\lambda_P$ and $\lambda_{RR}$, describe the effect that parental cells have on radioresistant cells, and vice versa; the interactions may be competitive ($\lambda_P > 0$ and $\lambda_{RR} > 0$), mutualistic ($\lambda_P < 0$ and $\lambda_{RR} < 0$) or antagonistic ($\lambda_P < 0 < \lambda_{RR}$ or $\lambda_{RR} < 0 < \lambda_P$)[11]. To confirm our ability to reliably infer the interaction parameters in our experimental data, we performed in silico studies in which the Lotka–Volterra model was used to generate synthetic growth curves of mixed spheroids with known interaction parameters ($\lambda_P = 0.5$ and $\lambda_{RR} = 0.5$) and three levels of experimental noise (5, 10, and 20%). Interaction parameters estimated by fitting only total volume measurements to the Lotka–Volterra model resulted in large uncertainties, whereas interaction parameters estimated using volume measurements supplemented with measurements of the ratio of the mixed populations decreased uncertainty in the parameter estimates (Supplementary Fig. 3). Thus, using both volume measurements and cellular proportions from our experimental data (Fig. 2A, B), we predicted competitive interactions in PC3 spheroids ($\lambda_P > 0$ and $\lambda_{RR} > 0$) and antagonistic interactions in DU145 spheroids ($\lambda_P > 0 > \lambda_{RR}$; Fig. 3B).

Experimental data supported the mathematical predictions, as measured by changes in the cell cycle duration of each population isolated from homogeneous spheroids compared to those isolated from mixed spheroids (Supplementary Fig. 4). In mixed PC3 spheroids, uptake of the nucleoside analogue 5-ethynyl-2-deoxyuridine (EdU$^+$) in parental and radioresistant populations on day 5 reduced by 25% (parental, $P_{adj} = 0.019$; RR, $P_{adj} = 0.007$; Fig. 1A); parental cells still had reduced EdU$^+$ uptake on day 10 ($P_{adj} < 0.001$; Fig. 3C). Although this competition during growth resulted in reduced proliferation for each population, it also led to reduced death for each population (Fig. 2C), thus enhancing overall viability. By contrast, in DU145 mixed spheroids, the parental population had a 50% increase in EdU$^+$ uptake on day 10 ($P_{adj} = 0.015$), whereas the radioresistant population had an 80% reduction in EdU$^+$ uptake on day 15 ($P_{adj} < 0.001$; Fig. 3C). The populations in DU145 spheroids thus behave antagonistically: the parental gains a growth advantage, while the radioresistant is adversely affected. Although the mechanism (i.e., competitive vs. antagonistic interactions) varies between the cell lines, these results support the premise that ecological-type interactions between heterogeneous tumour populations enhance tumour growth in the absence of radiation.

**Mixed cultures of tumour cell populations have reduced radiation sensitivity.** To determine whether mixed cultures had altered radiotherapy response, we subsequently irradiated

homogeneous and mixed PC3 spheroids at a range of doses and measured the time of growth to the endpoint, as assessed by radiation-induced growth delay. At each dose of radiation, growth delay curves differed significantly (on a global scale) among the four spheroid groups (parental, 9:1 parental:RR, 1:1 parental:RR, and RR). We subsequently tested whether the inclusion of 10% radioresistant cells could reduce growth delay. Compared to PC3 spheroids comprising 100% parental cells, mixed spheroids (seeded 9:1 parental:RR) had significantly decreased growth delay after irradiation at 7.5 Gy ($P_{adj} = 0.005$; Fig. 4A) and 10 Gy ($P_{adj} = 0.02$; Fig. 4A). In a separate experiment using only 6 Gy radiation, the growth delay after radiation was reduced in mixed PC3 spheroids (seeded 1:1; $P_{adj} < 0.001$; Fig. 4B), although radioresistant cells likely drove regrowth because they comprised $61 \pm 5.6\%$ and $79 \pm 5.6\%$ of cells in mixed spheroids on days 10 and 15, respectively (Fig. 4C). Relative to parental cells isolated from homogenous spheroids, those isolated from mixed spheroids on days 10 and 15 had a >22% reduction in cell death ($P_{adj} = 0.045$; Fig. 4D). However, EdU$^+$ uptake decreased by 46% and the proportion of cells in G$_2$M/EdU$^-$ increased ($P_{adj} < 0.001$; Supplementary Fig. 5), suggesting arrest in these cell cycle phases.

Similar results were obtained for mixed DU145 spheroids. After irradiation at 6 and 10 Gy, the growth delay for mixed spheroids was reduced compared to parental spheroids (6 Gy, $P_{adj} = 0.001$; 10 Gy, $P_{adj} < 0.001$; Fig. 4E). Regrowth at 6 Gy was not due to the radioresistant population dominating, as the parental population comprised >60% of the mixed spheroids (Fig. 4F). Furthermore, relative to the parental population in homogeneous spheroids, those in mixed spheroids had a 40% decrease in cell death on day 10 ($P_{adj} = 0.045$; Fig. 4G) but a cell cycle arrest (as evidenced by a 19% reduced EdU$^+$ uptake and an increase in G$_2$M/EdU$^-$ phases ($P_{adj} = 0.045$; Supplementary Fig. 5)). In contrast, compared to the radioresistant population from homogeneous spheroids, those from mixed spheroids had a 37% increase in cell death on day 15 ($P_{adj} = 0.038$; Fig. 4G), but no significant changes in EdU$^+$ uptake or in G$_2$M/EdU$^-$ proportions (Supplementary Fig. 5). Together, these data indicate that radioresistant populations not only reduce the growth delay upon irradiation but also enhance the survival of the parental population in the irradiated mixed spheroids.

**The type of ecological interaction dictates regrowth time after radiation.** We then investigated whether the type of ecological interaction that occurs could impact the regrowth time after radiotherapy. Using the Lotka–Volterra model, we simulated regrowth of mixed spheroids (seeded 1:1 in silico) after exposure to 6 Gy for a range of values of the interaction parameters $\lambda_P$ and $\lambda_{RR}$. As expected, competitive cell populations ($\lambda_P < 0$ and $\lambda_{RR} < 0$) have the slowest regrowth times (50–200 days), while those with mutualistic interactions ($\lambda_P > 0$ and $\lambda_{RR} > 0$) have the fastest regrowth times (14–32 days) (Fig. 5A–D). Interestingly, within a specific interaction type, the *values* of the interaction parameters $\lambda_P$ and $\lambda_{RR}$ impacted the regrowth time. For example, for competitive interactions, combinations of values of $\lambda_P$ and $\lambda_{RR}$

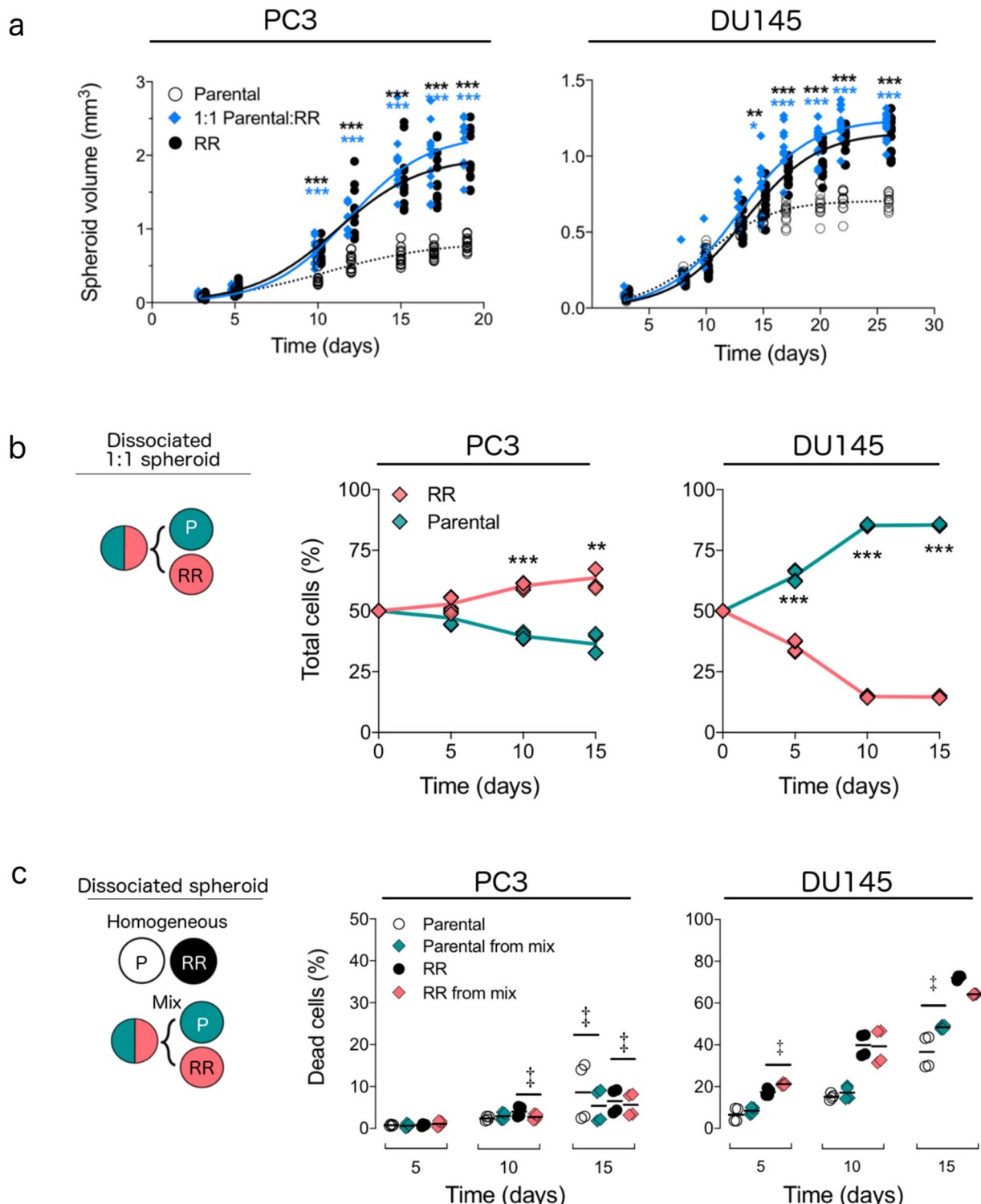

**Fig. 2 Inclusion of radioresistant cell populations enhances the growth of unirradiated mixed 3D cultures and alters the survival of each population.**
**a** Growth curves of 3D spheroids comprising parental (white circles), radioresistant (RR, black circles), or 1:1 mix (blue diamonds) of parental:RR populations ($n = 12$ spheroids for all except $n = 11$ for DU145 mix, pooled from 2 batches). Lines are the logistic growth model fits. $*P_{adj} < 0.05$, $**P_{adj} < 0.01$, $***P_{adj} < 0.001$, 2-way ANOVA with post hoc Bonferroni correction comparing parental vs. mixed (blue asterisks) and parental vs. RR (black asterisks) spheroids. **b** Proportions of fluorescent parental (GFP, green diamonds) and radioresistant (DsRed, pink diamonds) populations measured over time using flow cytometry from spheroids seeded as 1:1 mixture ($n = 4$ experiments). Lines connecting symbols indicate the median value. $***P_{adj} < 0.001$, 2-way ANOVA with Sidak correction. **c** Percent of dead cells in each population isolated from homogeneous spheroids (parental, white circles; RR, black circles) or from mixed spheroids (parental, green diamonds; RR, pink diamonds) seeded as a 1:1 mixture of parental:RR ($n = 4$ experiments). Bar indicates mean value. ‡FDR < 0.05, overdispersed binomial regression with Benjamini–Hochberg correction. No $P$ value was calculated for RR populations from DU145 spheroids on day 15 due to a poor statistical model fit.

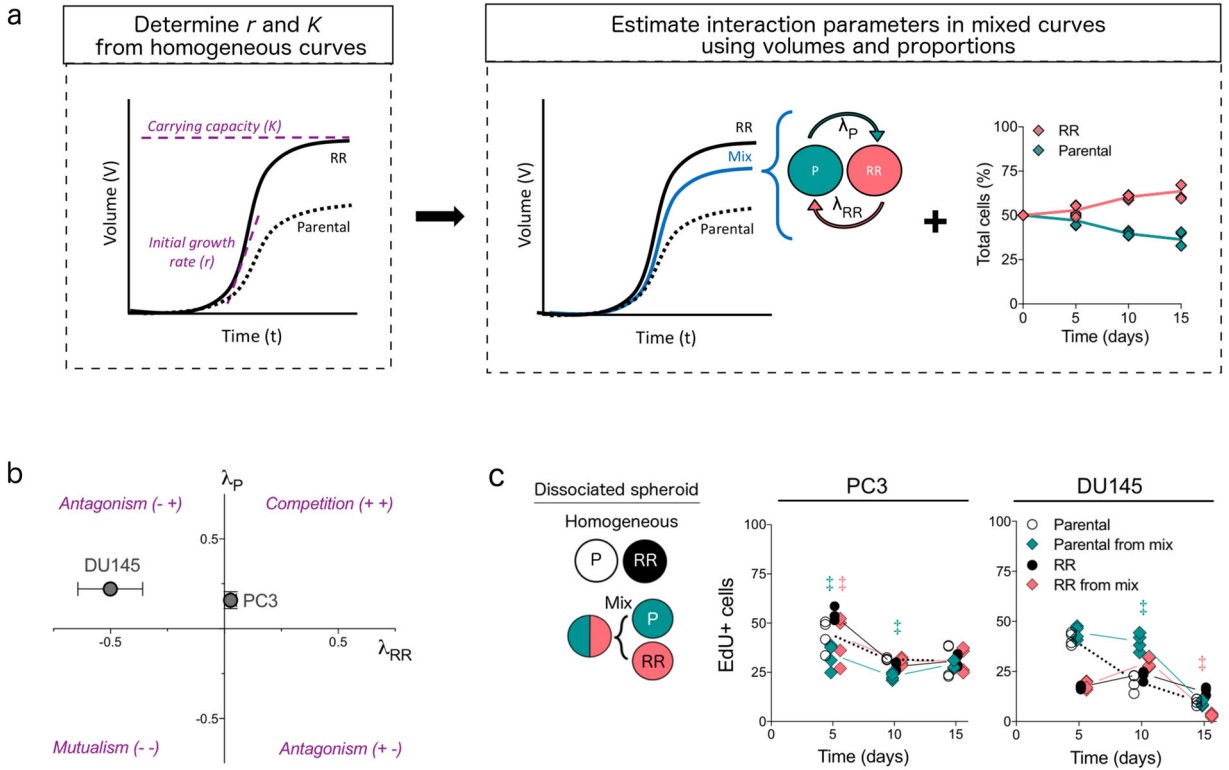

**Fig. 3 Mathematical modelling predicts the type of ecological interactions in unirradiated mixed prostate spheroids. a** Schematic illustrating pipeline for mathematical modelling of ecological interactions. The logistic model of spheroid growth was fit to growth curves of homogeneous PC3 and DU145 spheroids to determine the growth rate $r$ and carrying capacity $K$ (the limiting volume of the spheroid) for parental and radioresistant populations. Interaction parameters in mixed curves were then estimated by fitting volume measurements and proportions of each population to a Lotka–Volterra model. The parameter $\lambda_P$ indicates the effect of parental on radioresistant, while the parameter $\lambda_{RR}$ indicates the effect of radioresistant on parental. **b** Estimated interaction parameters using biological data of mixed PC3 and DU145 spheroids. The nature of the interaction, i.e., competition vs. antagonism, depends on the sign of the interaction parameters. Error bars represent confidence intervals. **c** Experimental determination of proliferation (EdU+ cells) from cell cycle analysis for parental and radioresistant (RR) cell populations isolated from mixed (parental, green diamonds; RR, pink diamonds) and homogeneous spheroids (parental, white circles; RR, black circles; $n = 4$ experiments). Lines connecting symbols indicate the median value. ‡FDR < 0.05, overdispersed Poisson regression model fitting with Benjamini–Hochberg correction.

**Table 1 Values of growth rate $r$, carrying capacity $K$, and initial volumes $V_0$ for parental and radioresistant homogeneous spheroids from PC3 and DU145 cell lines.**

| | Cell line | |
|---|---|---|
| **Parameter** | **PC3** | **DU145** |
| $r_P$ | 0.293 (0.263, 0.323) | 0.306 (0.280, 0.332) |
| $r_{RR}$ | 0.363 (0.339, 0.386) | 0.210 (0.196, 0.224) |
| $K_P$ | 0.843 (0.754, 0.933.) | 0.724 (0.686, 0.762) |
| $K_{RR}$ | 2.217 (2.000, 2.433) | 1.388 (1.267, 1.508) |
| $V_P(0)$ | 0.040 (0.033, 0.047) | 0.030 (0.025, 0.036) |
| $V_{RR}(0)$ | 0.036 (0.032, 0.042) | 0.046 (0.041, 0.052) |

Values indicate mean value and 95% confidence intervals (in italics).
P parental, RR radioresistant.

between 0.5 and 0.75 day$^{-1}$ mm$^{-1}$ generate regrowth in 100–200 days (sharp peaks), while those ranging between 0 and 0.5 day$^{-1}$ mm$^{-1}$ generate regrowth in about 50 days (Fig. 5B). Furthermore, for antagonistic interactions, when $\lambda_P < 0 < \lambda_{RR}$ regrowth is slower than when $\lambda_P > 0 > \lambda_{RR}$ by a factor of almost 2 (Fig. 5A, D). These results suggest that the type of ecological interaction influences the time of regrowth after radiotherapy and that it could contribute to variability in tumour responses to radiotherapy.

**Oxygen constraints and cell–cell communication mediate enhanced growth and reduced radiation sensitivity of mixed PC3 spheroids.** Having found that ecological interactions enhance the growth and decrease radiation sensitivity of mixed tumour populations, we subsequently tried to determine whether microenvironmental pressures and/or cell–cell communication could mediate the observed phenotypes. We focused in the first instance on mixed PC3 spheroids, which showed enhanced survival and reduced radiation sensitivity despite having competitive interactions. Since microenvironmental pressures, such as spatial constraints, affect cellular interactions in chemotherapy[13], we evaluated the spatial distribution of each population in homogeneous and mixed spheroids using microscopy. Unlike those isolated on day 5, sections of mixed PC3 spheroids isolated on day 11 showed parental cells located predominantly in the centre, a region that overlapped with staining for hypoxia, but not for proliferation. Spheroids comprising mixed and radioresistant cells were visually less compact (Fig. 6A) than those comprising parental cells. The localisation of parental cells in mixed spheroids, which was detected in ~70% of spheroids (Supplementary Fig. 6), was also coupled with a decrease in radiation sensitivity. The clonogenic survival of the parental population isolated from untreated mixed spheroids on day 11 was 25% higher ($P_{adj} = 0.04$) than that of parentals isolated from homogeneous spheroids; this effect was not seen in populations isolated from day 5 spheroids (Fig. 6B).

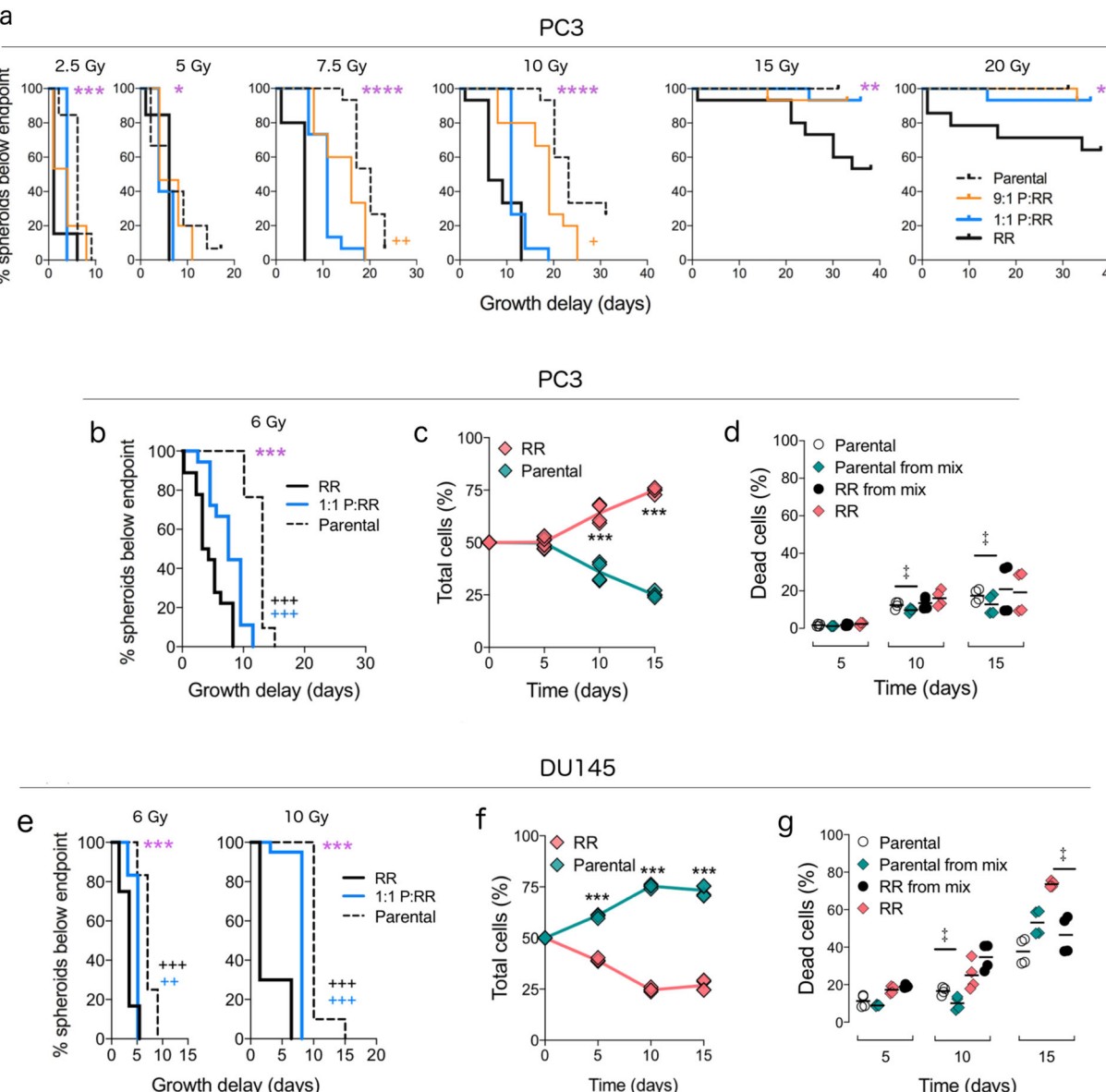

**Fig. 4 Inclusion of radioresistant populations in spheroids reduces radiation sensitivity. a** Survival curves of time (days) taken for irradiated PC3 spheroids seeded with 100% parental (dashed black line), 9:1 parental:radioresistant (P:RR, orange line), 1:1P:RR (blue line) or 100% RR (black line) cells to reach endpoint, adjusted for the time taken by untreated spheroids to reach the same endpoint ($n = 15$ spheroids/group/radiation dose from 1 batch). ***$P_{adj} < 0.001$, global log-rank test for all survival curves; +$P_{adj} < 0.05$, ++$P_{adj} < 0.01$, log-rank test comparing survival curves of parental vs. 9:1P:RR (orange crosses). **b**, **e** Survival curves of time (days) taken for irradiated PC3 (6 Gy) and DU145 (6 and 10 Gy) spheroids seeded as parental (dashed black line), 1:1P:RR (blue line), or RR (solid black line) to reach the endpoint, adjusted for the time taken by untreated spheroids to reach the same endpoint ($n = 17$ spheroids/group for PC3 pooled from 3 batches; $n = 12$ spheroids/group for 6 Gy DU145 spheroids pooled from 2 batches; $n = 20$ spheroids/group for 10 Gy DU145 spheroids from 1 batch). ***$P_{adj} < 0.001$, global log-rank test for all survival curves; ++$P_{adj} < 0.01$, +++$P_{adj} < 0.001$, log-rank test comparing parental vs. 1:1P:RR (blue crosses) and parental vs. RR (black crosses) spheroids. **c**, **f** Proportions of parental (green diamonds) and RR (pink diamonds) populations measured over time using flow cytometry from spheroids seeded as 1:1 mixture and irradiated ($n = 4$ experiments, ***$P_{adj} < 0.001$, 2-way ANOVA with Sidak correction). Lines connecting symbols indicate the median value. **d**, **g** Percent of dead cells in each population isolated from irradiated homogeneous spheroids (parental, white circles; RR, black circles) or from mixed spheroids (parental, green diamonds; RR, pink diamonds) seeded as a 1:1 mixture of parental:RR ($n = 4$ experiments). Bar indicates mean value. ‡FDR < 0.05, overdispersed binomial regression with Benjamini–Hochberg correction. No $P$ value was calculated for parental populations from DU145 spheroids on day 15 due to a poor model fit.

Based on these results, we hypothesised that the spatial structure may result from competition for oxygen or differences in oxygen consumption rate (OCR) between the two cell populations. To test this hypothesis, we developed a cellular automaton (CA) model describing the changes in the size and structure of a 2D cross-section through a 3D tumour spheroid in which cells from parental and radioresistant populations divide and die in response to the local oxygen concentration. Parameter values for proliferation were derived from our flow cytometry data; measured OCR values were 17% lower for radioresistant than for parental cells ($P = 0.001$, Fig. 6C). The model was initialised by seeding equal numbers of parental and radio-resistant cells at day 0, based on the estimated values of $V_P(0)$ and $V_{RR}(0)$ from the logistic model. On day 10 of our in silico

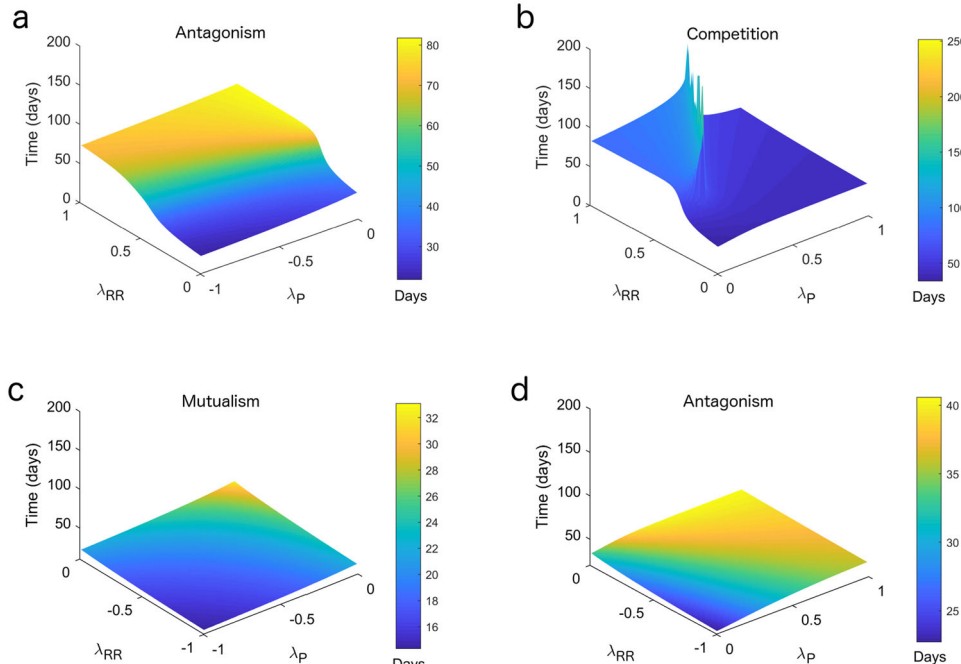

**Fig. 5 The type of ecological interaction alters regrowth time after radiation. a–d** For each type of ecological interaction (i.e., antagonism, competition, and mutualism), mixed spheroids were grown in silico using a range of interaction parameters $\lambda_P$ and $\lambda_{RR}$ until the volume reached 0.9 mm³. At this volume, spheroids were irradiated (6 Gy) in silico and monitored for regrowth until they reached 0.9 mm³. Radiation damage was simulated using the linear-quadratic model, for which input values ($\alpha$ and $\beta$) were determined from experimental PC3 clonogenic assays: parental population, $\alpha = 0.44 \pm 0.07$ and $\beta = 0.04 \pm 0.01$; radioresistant population, $\alpha = 0.35 \pm 0.06$ and $\beta = 0.03 \pm 0.01$. The numbers of parental and radioresistant cells at the time of radiation depend on the values of $\lambda_P$ and $\lambda_{RR}$, whereas the numbers post-radiation depend on the effect of radiation on each population, and on the values of $\lambda_P$ and $\lambda_{RR}$. See Supplementary Methods section 1 for more details on the mathematical model.

simulations, the radioresistant cells were located close to the spheroid periphery, while parental cells were located close to the centre (Fig. 6D). We then used the CA model to determine whether the growth curves of mixed spheroids could be explained by competition for oxygen. Although the CA model predicted the growth curves of the homogeneous spheroids, the model did not accurately capture the growth curves of the mixed spheroids under the assumption that the two cell populations compete for oxygen (Fig. 6E). Thus, although competition for oxygen may explain the spatial structure of cell populations and the decreased radiation sensitivity of mixed spheroids, OCR differences alone are not sufficient to explain their growth phenotype.

We subsequently looked for evidence of cell–cell communication between parental and radioresistant cells in hypoxia, given that hypoxia appeared to be an important factor for enhanced survival. Cells were co-cultured in hypoxia for 120 h using Transwell inserts; in this system, cells can share factors (e.g., exosomes, microRNA and proteins) but are not in direct contact with each other. The number of parental cells co-cultured with radioresistant cells was 48% higher than the number measured from homogeneous cultures in hypoxia but not in normoxia ($P_{adj}$ < 0.001; Fig. 6F). Altogether, our results indicate that both microenvironmental pressures (through oxygen constraints), and cell-cell communication (through the transfer of factor(s)) mediate ecological interactions between cell populations that can result in enhanced growth and reduced radiation sensitivity of mixed PC3 spheroids.

## Discussion
We evaluated whether ecological interactions between tumour cell populations with different radiation sensitivities alter tumour growth kinetics and radiation response. Using biological experiments and ecological mathematical models, we found that

prostate cancer spheroids comprising mixed tumour cell populations have enhanced growth and reduced radiation sensitivity. Numerical simulations predicted that these phenotypic changes result from competitive and antagonistic interactions between the cell populations in spheroids, and result in variable regrowth times after radiation. By demonstrating that ecological-type interactions impact growth and radiation response, our study presents a new aspect that could be considered in radiobiological models to target ITH and reduce local recurrence in prostate cancer.

Ecological interactions between tumour cell populations, irrespective of the type of interaction, appear to have evolutionary advantages for cancer. We found that both competitive and antagonistic interactions increase bulk tumour growth and resistance to radiation. Our results are consistent with findings from previous studies investigating the impact of ecological interactions in chemotherapy: although the *type* of interaction (e.g., competitive vs. mutualism) can vary, ecological interactions appear to increase resistance to cancer therapy[14,17,27–29]. Furthermore, in our study, interactions increase the growth and post-radiation survival for the parental population in mixed spheroids in both cell lines. The interactions also did not result in the elimination of either population, recapitulating the multiclonality observed in human prostate cancer[3,4,22,30,31]. Current radiotherapy methods used to target more "resistant" populations, such as dose escalation, may therefore not be sufficient to target ITH if interactions between tumour cell populations reduce overall tumour sensitivity to radiation while maintaining ITH. Future studies could test whether accounting for these interactions in radiobiological models improves tumour response to radiation.

We also found that interactions mediated by oxygen constraints alter the spatial distribution of prostate cell populations in

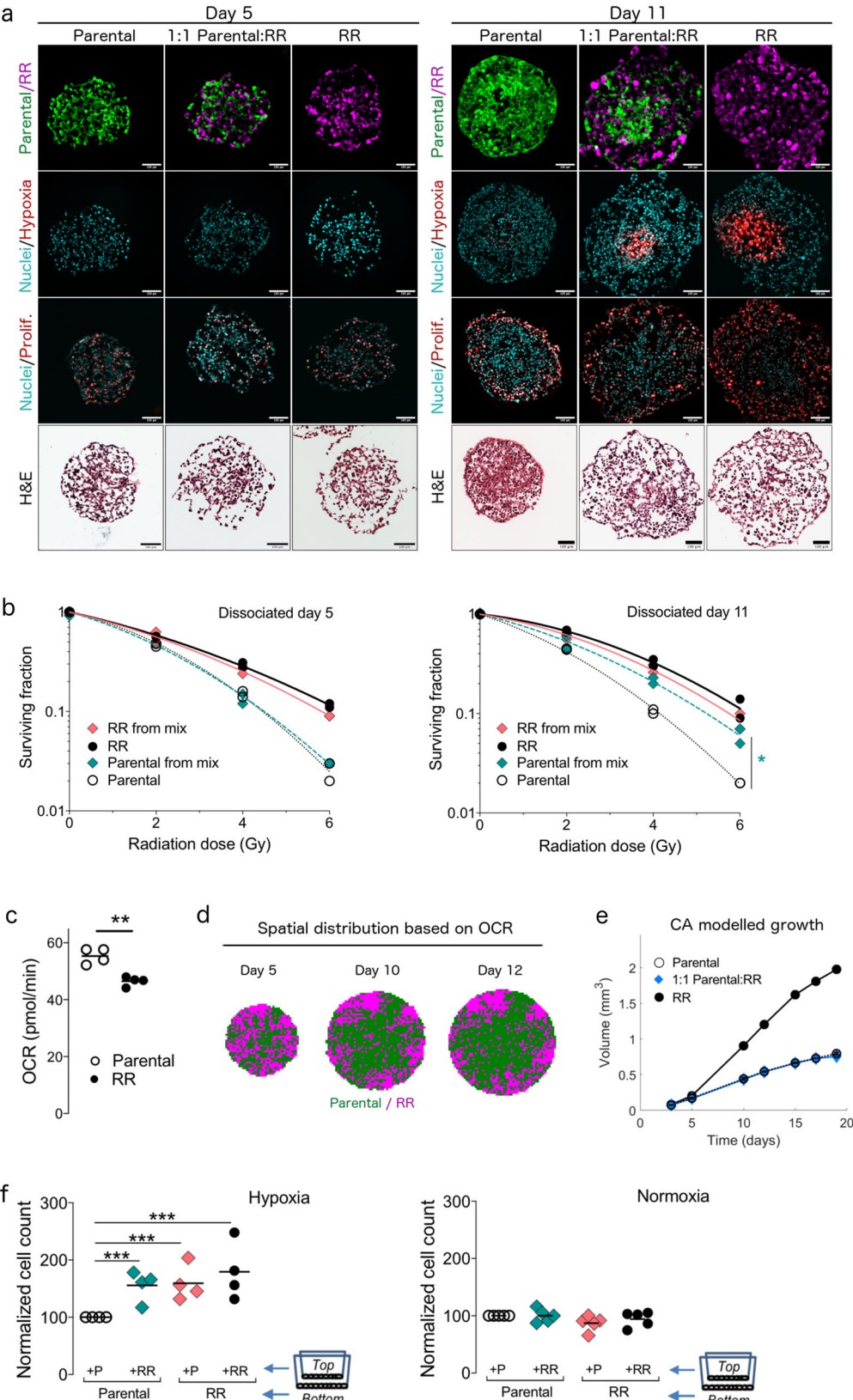

mixed PC3 spheroids. Both our biological and spatially explicit mathematical models showed radioresistant cells in the periphery and parental cells in hypoxic regions. These findings are consistent with previous reports showing that the metabolic demands of drug-resistant cancer cells affect their spatial localisation[7]. The observed spatial heterogeneity in our study is important because

radioresistant cells located in the periphery could metastasise[16], while parental cells in hypoxia would have reduced sensitivity to radiation. Since the efficacy of radiotherapy may also depend on the spatial distribution of tumour populations, these results suggest that spatially-explicit mathematical models may be needed to further optimise radiation treatments. For example, the

**Fig. 6 Oxygen competition and cellular cross-talk mediate enhanced growth and reduced radiation sensitivity in unirradiated mixed PC3 spheroids. a** Images showing spatial localisation of parental (green pseudo-colour) and radioresistant (RR, magenta pseudo-colour) cell populations, markers of nuclei (Hoechst, cyan pseudo-colour), hypoxia (EF5, red pseudo-colour) and proliferation (Ki67, red pseudo-colour), and H/E staining in homogeneous and mixed PC3 spheroids isolated on days 5 and 11. To indicate overlapping areas, the six spheroid sections shown in the row spatial localisation are also shown in the row nuclei/hypoxia, with different channels merged together as indicated by row title. Spheroid sections shown in row nuclei/proliferation are different. Scale bar, 100 μm. **b** Clonogenic survival of cell populations isolated from untreated spheroids dissociated at day 5 and day 11 ($n = 2$ experiments). *$P_{adj} = 0.04$, 1-way ANOVA with Sidak correction. **c** Oxygen consumption rate (OCR) of parental (white circles) and RR (black circles) PC3 populations, normalised to cell density ($n = 4$ experiments). Bar indicates mean value. *$P_{adj} < 0.01$, paired $t$ test, two-tailed. **d** The spatial distribution of cell populations based on OCR in mixed PC3 spheroids, as predicted by the cellular automaton (CA) model. **e** Growth curves predicted solely by competition for space and oxygen in untreated homogeneous (parental, white circles; RR, black circles) and mixed (blue diamonds) spheroids. The CA model described in d was used to generate a growth curve (see Supplementary Methods section 2). **f** Cell count of parental and RR cells cultured separately or together in hypoxia (0.1% $O_2$) or in normoxia using Transwell inserts, in which cells share medium but are separated by a membrane ($n = 4$ experiments for hypoxia; $n = 5$ experiments for normoxia). Bar indicates mean value. *$P_{adj} < 0.01$, **$P_{adj} < 0.001$, using overdispersed Poisson regression.

delivery of non-uniform doses of radiation based on the spatial expression of prostate cancer antigen has been shown to improve the tumour control probability in prostate cancer[32].

Since experimental approaches to studying ecological interactions can be complex, time-consuming, and/or have limited data, interdisciplinary approaches that treat tumours as evolving ecosystems may be necessary to target ITH[10] and reduce local recurrence in prostate cancer. Mathematical simulations, with input data from experiments, can help evaluate the outcome of different treatment strategies and test biological hypotheses. The success of this approach was recently demonstrated in a pilot clinical trial for patients with drug-resistant prostate cancer: tumour population dynamics from each patient were used to inform a Lotka–Volterra model and optimise a chemotherapy regimen that significantly delayed recurrence than standard-of-care[18]. In our study, we used mathematical models to guide the experimental design of flow cytometry experiments, as well as to test whether oxygen constraints could predict enhanced growth of mixed spheroids. Conversely, we used experimental data to infer parameters with tighter confidence intervals in theoretical models, which refined the ability of our mathematical models to predict ecological interactions. Our approach can be applied to evaluate the impact of interactions during and following radiotherapy in other cancer types and systems.

Although relatively simple, our modelling approach has three advantages over previous approaches. First, unlike other ecological models used to study cellular interactions that only model competition between sensitive and resistant populations[15,18,33,34], our Lotka–Volterra model describes a range of interactions, including competition, mutualism, and antagonism. Second, although the model does not account for different cell states (e.g., hypoxic, proliferative and necrotic) or for the nature of the cell–cell interactions to change over time, it accurately predicted the dominant interactions occurring within spheroids. Third, the small number of parameters in the mathematical model can be easily estimated from and validated by experimental data. One limitation is that our Lotka–Volterra model did not account for possible changes in the cellular interactions over time, which might occur due to changes in the tumour microenvironment. Since tumour populations can evolve over time, future studies are needed to investigate whether changing interactions (especially after treatment) could affect regrowth. While our spatially-resolved hybrid CA model was sufficient to study the impact of the microenvironment (i.e., oxygen constraints) on tumour development in a 2D cross-section of a tumour spheroid, the model in its current form also does not account for direct interactions between cell populations. To generate more realistic predictions, the model could be extended to three-dimensional space, along with rules governing cell behaviour that are modified to account for interactions between different cell populations.

Since cells can interact directly and indirectly via secretion of molecules, our model could be further modified to explicitly track factors released by coexisting cell populations. These modifications, whilst adding complexity to the model, would allow us to test various hypotheses pertaining to the impact of ITH on tumour growth and dynamics. Finally, while neither model accounts for angiogenesis, which would occur in vivo, both models could be extended to account for the role of the vasculature in tumour response to radiotherapy, by, for example, exploiting existing models of vascular tumour growth[35].

## Methods

### Biological experiments

*Cell culture.* We obtained unlabelled parental (sensitive) and radiation-resistant populations from two prostate cell lines, PC3 and DU145, from the Liu laboratory (University of Toronto, Canada). Radioresistant cell populations comprised pooled cells from the parental line that survived a clinically-relevant course of radiotherapy[23,24]. To produce stable, fluorescent cell lines, we transduced cells using lentiviral particles containing the vectors, pCDH1-CMV-GFP-EF1-Hygro or pCHD1-CMV-DsRed-EF1-Hygro (Systems Biosciences), collected the top 30% of brightest cells by flow cytometry, and used hygromycin B (50 mg/mL, Gibco) for selection (200 μg/mL for PC3 and 250 μg/mL for DU145). Cell lines were cultured in DMEM medium (low glucose, pyruvate, GlutaMAX, Gibco) supplemented with 25 mM HEPES (Gibco), 10% foetal bovine serum (Sigma or Pan-Biotech), and 1% penicillin/streptomycin. Authentication was performed using STR profiling (Promega PowerPlex 21 PCR kit, Eurofins), and mycoplasma checks were performed routinely using MycoAlert Mycoplasma Detection Kit (Lonza). All cells were maintained in an incubator (37 °C, 5% $CO_2$). Unlabelled cells were cultured for up to 10 passages (~6 weeks) for transduction; labelled cells were cultured for up to 10 passages (~6 weeks). No additional courses of radiation were used to maintain resistance. We measured clonogenic survival at 2 Gy (SF2) to verify that the labelled cells maintained the resistance phenotype for up to 12 passages (~8 weeks) in culture.

*Monolayer growth experiments.* Single cells ($1 \times 10^3$ cells/well, 200 μL medium) were seeded in triplicate in flat-bottom 96-well plates, allowed to attach overnight, irradiated (0, 2, or 6 Gy), and imaged daily using brightfield (Incucyte Live Cell Imaging System, Sartorius). The medium was changed every 2 days. Cell confluence was determined using Incucyte Base Software (Sartorius).

*Clonogenic assays.* Survival of cell lines after radiation was measured using a clonogenic assay[36]. Briefly, cells were seeded in triplicate in six-well plates and irradiated using a Cs-137 (dose rate of 0.89 Gy/min) or an X-ray irradiator (195 kV, 10 mA). Surviving colonies were stained after 10 days with crystal violet and counted. The surviving fraction was calculated as (number of colonies/number of seeded cells) × plating efficiency.

*Response to cisplatin.* To check whether the RR cells had altered DNA damage response, we measured the cell viability of each population in response to a range of concentrations of cisplatin using a modified cytotoxicity assay[37]. Briefly, single cells ($2 \times 10^3$ cells in 100 μL/well) were seeded as monolayers in triplicate in a 96-well plate and allowed to attach for 36 h before treatment. Increasing concentrations of cisplatin (made in 100 μL/well) were added to each well resulting in a final volume of 200 μL/well; cisplatin (Sigma) was prepared fresh for each treatment by dissolving the powder in 0.9% sterile-filtered saline to a stock concentration of 3.3 mM. Treated cells were then cultured for 72 h in cisplatin before they were fixed with 10% formalin. Cell confluence was determined using Incucyte Base Software

(Sartorius) and reformatted to a concentration–response curve by normalising cell confluence values to the untreated well.

**Spheroid generation and culture.** Homogeneous and mixed spheroids were generated in 96-well, ultra-low attachment plates (7007, Corning) by seeding different ratios of parental and radioresistant cell populations ($2 \times 10^3$ total cells/well) using Matrigel (5% v/v, Corning) to promote spheroid formation[38]. For all spheroid experiments, after a formation phase of 3 days, spheroids were fed every 2 days by replacing 50% of the medium in each well with fresh medium (200 µL total/well). Culture medium and incubation conditions were as described under 'Cell culture'. Spheroid volumes were calculated using SpheroidSizer[39].

**Unirradiated spheroid growth experiments.** Spheroids were generated as described in 'Spheroid generation and culture' and monitored for growth by brightfield imaging (Leica DM IRBE, Hamamatsu).

**Flow cytometry.** The proportions, survival, and cell cycle of each population from spheroids were measured by flow cytometry. Mixed unirradiated or irradiated spheroids (seeded 1:1 parental:RR, 6–8 pooled/group) were incubated with EdU (10 µM final concentration) 12 h prior to dissociation, dissociated (100 µL Accumax, Millipore) for 20 min at 37 °C, washed with phosphate-buffered saline (PBS), centrifuged (300 × g, 5 min), and incubated with efluor-780 (1 µL/mL PBS; ThermoFisher Scientific) for 30 min on ice in the dark to distinguish live/dead cells. After washing in PBS, samples were fixed for 10 min in IC Fixation Buffer (ThermoFisher Scientific), and permeabilised and stained with Click-iT Plus EdU Alexa Fluor 647 (ThermoFisher Scientific) according to manufacturer's instructions. Following a wash in 1× saponin, cells were incubated 30 min with FxCycle Violet Stain (1:1000, 300 µL of 1× saponin; ThermoFisher Scientific) before being run on the BD LSR Fortessa X-20 Cytometer or the Attune NxT Flow Cytometer using the 405, 488, 561, and 633 lasers. Data were analysed using FlowJo (Treestar, Inc.) as described in Supplementary Figs. 2 and 4.

**Spheroid growth experiments after radiation.** To determine bulk radiation response of spheroids, PC3 cells were seeded as spheroids ($n = 15$ per dose per group) with 4 groups as described in 'Spheroid generation and culture': parental, 9:1 parental:RR, 1:1 parental:RR, and RR. After formation, spheroids were irradiated (0, 2.5, 5, 7.5, 10, 15, and 20 Gy) on day 4 and imaged for up to 48 days to monitor regrowth using brightfield (Celigo Imaging Cytometer, Nexelcom). After log-transforming the volume data, we calculated the radiation-induced growth delay (days) relative to untreated spheroids as the time for each irradiated spheroid to reach a volume endpoint (2.5 times the starting volume right before irradiation); we selected the lowest endpoint that was still within the exponential growth phase of all spheroids in the experiment. The average time for untreated spheroids to reach endpoint was estimated in R by local regression using the *loess* function with "direct" surface estimation to allow extrapolation for the parental spheroids (R project, v. 3.6.2).

Regrowth experiments were repeated by irradiating day 3 spheroids from three groups (parental, mixed and RR) of PC3 cells (6 Gy, $n = 17$–18/ group) and of DU145 cells (6 Gy, $n = 12$/group; 10 Gy, $n = 20$/group). Spheroids were imaged using brightfield (Leica DM IRBE, Hamamatsu) for up to 27 days (PC3) and up to 23 days (DU145). The radiation-induced growth delay (delays) was calculated as above, but with different endpoints (3.5 times starting volume for PC3 and 4 times starting volume for DU145) to ensure the endpoint was within the exponential growth phase. Data from Fig. 2 were used to estimate average time of untreated spheroids; the 'span' parameter of the *loess* function was reduced from the default of 0.75–0.5 for the unirradiated DU145 spheroids to better estimate the average time of reaching the endpoint.

To measure changes in the radiation response of PC3 cell populations within spheroids, untreated homogeneous and mixed spheroids were grown until day 5 or 11, dissociated using Accumax, seeded as single cells for clonogenic experiments, and allowed to attach for 6 h prior to radiation. Fluorescent colonies were counted using the Celigo Cytometer.

**Immunofluorescence.** For immunofluorescence and H&E experiments, spheroids were treated and fixed prior to staining[40]. To investigate the spatial distribution of fluorescent populations, sections were hydrated in PBS, stained for 10 min with Hoechst (1 µg/mL in PBS, Sigma) to visualise nuclei, and mounted using ProLong Diamond Antifade Mountant (ThermoFisher). For hypoxia, spheroids were pre-treated with 300 µM of the hypoxia drug EF5 (gift from Dr. Cameron Koch, University of Pennsylvania) prior to fixation. They were then permeabilized (PBS containing 0.3% Tween-20, 10 min), blocked (5% goat serum in PBS containing 0.1% Tween-20, 30 min), stained using anti-EF5 antibody (75 µg/mL; from Dr. Cameron Koch) overnight at 4 °C, washed (ice-cold PBS containing 0.3% Tween-20, 2 × 45 min)[40], stained for nuclei as above, and mounted. For Ki67, spheroid sections were permeabilized (PBS containing 0.3% Tween-20, 10 min), blocked (5% goat serum in PBS containing 0.1% Tween-20, 30 min), and incubated overnight at 4 °C with primary antibody (clone SP6, 1:100, Vector Laboratories). After washing in PBS, sections were incubated for 1 h with goat anti-rabbit Alexa Fluor 647 (4 µg/ mL, ThermoFisher), washed, and stained with Hoechst 33342 (5 µg/mL, Sigma) for 10 min. Slides were mounted and imaged using epifluorescence microscopy (20× objective; 0.30 NA; 0.64 µm resolution; excitation lasers: 395, 470, 555, and 640;

Nikon Ti-E). Sections were stained using H&E and imaged using a Bright Field Slide Scanner (Aperio CS2, Leica) to visualise necrosis. To quantify the ratio of parental to radioresistant populations in spheroid cross-sections ($n = 16$ spheroids from 4 batches), we measured the number of pixels from each population (i.e., signal) by applying a threshold value 5 times higher than the median value of the background (i.e., noise) (Octave 4.4.1).

**Oxygen consumption measurements.** OCR was measured from each population using the Seahorse assay. Cells ($1.2 \times 10^4$/well) were seeded in triplicate using the normal culture medium in a Seahorse XF 96-well microplate (Agilent) and allowed to attach overnight. Prior to the assay, cells were washed with and incubated in assay medium (DMEM basal medium containing 5 mM glucose, 4 mM glutamine, 5 mM pyruvate, pH 7.4; 200 µL/well) for 2 h at 37 °C without $CO_2$ to degas the medium. Calibrant buffer (200 µL/well) was added to wells of the probe plate and also left at 37 °C without $CO_2$ to degas. After OCR was measured on the Seahorse XF Analyser (Agilent Biosciences), cells were fixed using 4% paraformaldehyde, stained using Hoechst 33342, and counted (Celigo Cytometer, Nexelcom).

**Transwell experiments.** Co-culture experiments were performed to measure whether transferred factors between cell populations enhanced survival under hypoxia. Cells were seeded in triplicate ($3.0 \times 10^4$/bottom well and $1.0 \times 10^4$/insert) in 12-well plates and in Transwell inserts, and allowed to attach overnight. Once the medium was changed, the plates were placed into normoxia or hypoxia (0.1% $O_2$) for 24 and 120 h. Cells were fixed using 4% paraformaldehyde, stained with Hoechst (5 µg/mL), and counted (Celigo Cytometer, Nexelcom).

**Statistics and reproducibility.** Data were evaluated for equal variance using homoscedasticity plots (absolute value of residual vs predicted value) and for normality using Q–Q plots (Prism 8.0, GraphPad). Unless otherwise indicated, statistical significance was evaluated using one-way ANOVA, two-way ANOVA, or a mixed-effects model followed by multiple testing correction ($\alpha = 0.05$). For clonogenic assays, the radiation protection factor was calculated as the area under the dose–response curve (AUC) for the RR cell populations divided by that of the parentals; AUC values were analysed for significance using a Student's $t$ test (unpaired, one-tailed, $\alpha = 0.05$). For cisplatin cytotoxicity assays, IC$_{50}$ values were calculated using a normalised response, variable slope, dose–response model (Prism, 8.0, GraphPad) and evaluated for statistical significance using extra sum-of-squares $F$-test. For post-radiation growth experiments, survival curves were analysed using the Mantel–Cox (log-rank) test and adjusted for multiple testing using Holm's correction; spheroids that did not reach the endpoint during the timeframe of the experiment were marked as 'censored' on the final day of the experiment (please see Supplementary Methods section 3 for further details). Due to heteroscedasticity, cell counts from flow cytometry experiments involving cell cycle and death, and from co-culture Transwell assays were analysed using over-dispersed Poisson or binomial regression models (please see Supplementary Methods section 3 for further details). For quantification of population proportions in microscopy images, pixel numbers were analysed using a two-tailed, Wilcoxon matched-pairs signed-rank test. Adjusted $P$ values ($P_{adj}$) are reported in the main text for experiments where multiple comparisons were performed.

Data points represent biological replicates; experiments were performed using at least two separate batches of cells. We note the following data exclusions: missing data from some time points due to technical failures in imaging (Fig. 1), one excluded mixed DU145 spheroid because its growth did not resemble that of the other 35 spheroids (Fig. 2a), and one excluded plate of PC3 spheroids from survival analysis (Fig. 4) because of irregular growth that did not match the other nine plates. Sample sizes were approximated using effect sizes from pilot studies to ensure power (approximate $\beta = 0.8$); randomisation and blinding were not possible.

## Mathematical experiments

**Non-spatial mathematical models.** We used the logistic growth model to describe the growth of homogeneous tumour spheroids[26]. Thus, the rate of change of spheroid volume $V$ at time $t$ is given by

$$\frac{dV}{dt} = rV\left(1 - \frac{V}{K}\right), \tag{1}$$

where $r > 0$ represents the growth rate, $K > 0$ is the carrying capacity (the limiting volume of the spheroid) and $V(t = 0) = V_0$ denotes the spheroid volume at $t = 0$. The analytical solution to the logistic model is given by

$$V(t) = \frac{V_0 K e^{rt}}{K + V_0(e^{rt} - 1)}. \tag{2}$$

The Lotka–Volterra model was used to describe the growth of mixtures of parental and RR cell populations

$$\left.\begin{aligned}
\frac{dV_P}{dt} &= r_P K_P \left(1 - \frac{V_P}{K_P} - \lambda_{RR}\frac{V_{RR}}{K_P}\right) \\
\frac{dV_{RR}}{dt} &= r_{RR} K_{RR} \left(1 - \frac{V_{RR}}{K_{RR}} - \lambda_P\frac{V_P}{K_{RR}}\right)
\end{aligned}\right\}, \tag{3}$$

with $V_P(t = 0) = V_{P0}$ and $V_{RR}(t = 0) = V_{RR0}$. In these equations, $V_P$ and $V_{RR}$

represent respectively the volumes of parental and RR populations, $r_P$ and $r_{RR}$ their initial growth rates, $K_P$ and $K_{RR}$ their carrying capacities, and $V_{P0}$ and $V_{RR0}$ their initial volumes. The parameters $\lambda_P$ and $\lambda_{RR}$ describe the effect that parental cells have on RR cells, and vice versa. These type of interactions, found in ecology[12], may be competitive ($\lambda_P > 0$ and $\lambda_{RR} > 0$), mutualistic ($\lambda_P < 0$ and $\lambda_{RR} < 0$) or antagonistic ($\lambda_P < 0 < \lambda_{RR}$ or $\lambda_{RR} < 0 < \lambda_P$)[41]. Further details, describing how the values of the model parameters were estimated, are included in Supplementary Methods section 1.

*Spatially resolved computational models.* To investigate the growth of heterogeneous spheroids we developed a spatially resolved CA model. Our CA model replicated the changes in the size and structure of a 2D cross-section through a 3D tumour spheroid suspended in the culture medium. The model couples a set of automaton elements arranged on a regular 2D grid to a partial differential equation describing the distribution of a growth-rate-limiting nutrient (here, oxygen) which is supplied from the culture medium surrounding the spheroid. Model simulations are initialised by placing a circular cluster of cells in the centre of the grid: this imitates seeding a spheroid in a Petri dish. Both cell populations consume oxygen that diffuses from the medium, and they proliferate and die at rates that depend on the local oxygen concentration and cell density. The parental and RR cells differ in cell cycle times, OCRs, sensitivity to changing oxygen concentration and rates of lysis. Further details about the implementation of the CA model are included in Supplementary Methods section 2.

*Modelling regrowth times after radiation.* Cell kill due to radiotherapy was incorporated into the models using the linear quadratic model[42], with α and β values derived for each cell population (PC3 parental and RR) from clonogenic survival curves. For the spatially resolved CA model, oxygen dependence of α and β was included; for the spatially averaged logistic and Lotka–Volterra models, oxygen dependence of α and β was neglected since oxygen levels are not included in these models.

**Reporting summary**. Further information on research design is available in the Nature Research Reporting Summary linked to this article.

## Data availability

The datasets that support the findings of this study are available in Zenodo with DOI: 10.5281/zenodo.4130692[43]. Any other relevant data are available upon reasonable request from the corresponding authors.

## Code availability

Computer code used to generate the surface plots in Fig. 5 (interactions_time_to_regrow), the cellular automaton model results in Fig. 6 (ca model), and the statistical analyses of cell cycle, post-radiation regrowth, and transwell data are available in Zenodo with DOI 10.5281/zenodo.4130692[43]. The source code implementing the cellular automaton model (ca_model) includes a demo on how to use it and images representing examples of expected outcomes. The computer code for mathematical modelling was written in MATLAB R2018b (v. 9.5) with Statistics and Machine Learning Toolbox (v.11.4), while the code for statistical analyses was written in R project (v. 3.6.2).

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

## Acknowledgements
We thank Dr. David Robert Grimes for helpful discussions, Dr. Yunhong Cao for technical assistance, and Dr. Graham Brown and Dr. Rhodri Wilson for assistance with microscopy (Microscopy Core Facility, Department of Oncology, University of Oxford). Access to the microscopes (Leica, Ti-E, and Celigo) and flow cytometer (Attune) were provided by the Microscopy and Flow Cytometry Core Facilities (Department of Oncology, University of Oxford). This research was supported by Cancer Research UK and the Engineering Physical Sciences Research Council (grant numbers: C5255/A12678, C2522/A10339, C53469/A19834, and C56606/A21440). BM, M Partridge, and PK were funded by the CRUK/EPSRC Oxford Cancer Imaging Centre. ALH was funded by Breast Cancer Now (grant number: 2015MayPR479). WWK was funded by KTH Royal Institute of Technology. M Paczkowski gratefully acknowledges the EPSRC for funding through a studentship at the Systems Biology programme of the University of Oxford's Doctoral Training Centre.

## Author contributions
Conception: A.L.H., M. Partridge, H.M.B., and P.K. Design: M. Paczkowski, L.K.S., A.L.H., M. Partridge, H.M.B., and P.K. Generation of cell lines: S.K.L. Experiments: M. Paczkowski, B.M., and P.K. Mathematical modelling: M. Paczkowski. Data analysis: M. Paczkowski, W.W.K., and P.K. Project supervision: A.L.H., M. Partridge, H.M.B., and P.K. Paper writing: M. Paczkowski, W.W.K., A.L.H., H.M.B., and P.K. All authors edited/reviewed the paper.

## Funding

## Competing interests
The authors declare no competing interests.
