## [Peer Review File · Communications Biology]

Reviewers' Comments:

Reviewer #1:

Remarks to the Author:

This is an outstanding investigation that seamlessly blends theoretical and empirical work in a way that allows the whole to be greater than the sum of its parts. It is an interesting and important subject. The manuscript is beautifully written. While I might quibble with some of the details and suggest some additional experiments, my overall recommendation is to publish it as is. My congratulations to the authors for excellent work.

Reviewer #2:

Remarks to the Author:

The study is generally fascinating and deploys multiple scientific disciplines to try and unravel how heterogeneity is implicated in resistance to radiation therapy in prostate cancer. At the same time, there are a lot of areas that need more explanation, significant under described methods, requires attention to detail, and other information that should be provided, buttressed or explained to provide better clarity. It's very challenging to read given the number of issues that came up and all the areas that cause confusion. In some areas I feel like the authors have created that confusion for themselves. I've enumerated a number of concerns here that are addressable but need attention. More specifically:

Resistance concerns:

- More information on the generation of these 'radioresistant' cell lines is needed. Were they generated in the lab as per reference 23 and 24? Or were they given to the lab from the originating authors? If it's the latter, what are the culture conditions required to maintain 'resistance', how often are they tested for 'resistance'? The resistance profiles are dissimilar between this paper and the reference, so some clarification would be useful.
- Biological characterization of the 'resistant' cells should be provided such that they can be compared to other evidence in the literature for 'radio resistance'.
- The data showing 'resistance' is completely underwhelming and not convincing, particularly S1B and C, with regards to generated resistance. Perhaps they should be called radio-treated? And in fact, in Suppl. Fig 6 it says parental and radio recurrent. Is RR radiation resistant or are these supposed to be radiation recurrent? Authors may want to consider calling these radio recurrent throughout because it's been confusing to contend with the data that they are indeed truly radioresistant.
- Why so many different ways of showing 'resistance' that tend to conflict with one another? For example, the clonogenic assays in Figure S1 stand in stark contrast to the data in figure 2a.
- Are these cell lines cross therapy resistant? This would be useful to test since, in relapsed and radiation refractory clinical cases, patients are sometimes treated with systemic therapies.
- DU145 appear to be more proliferative without radiation (S1C), can the treatment with 6gy be normalized to their baseline growth in order to conclude on proliferation rates after treatment? Perhaps then they won't be significantly different. This causes concern for the contention that they are 'resistant'

Other issues:

- Figure 2c the data indicate that du145 RR cells die at a rate of 20-40% every 5 days in monotypic culture, but they tend to grow in other experiments. Why? Also, it's perplexing that the monotypic and heterotypic cultures are insignificantly different in this panel even though the other data that are clearly less different are shown as statistically significant. How is that possible?
- The flow of data is a bit confusing -- the growth dynamics in the absence of radiation pressure

should be addressed first and then the mixed culture experiments with radiation would make more sense to bring in. Otherwise, it's like we're constantly going back and forth and confusing for the reader.

- What is the difference between S1B and Fig 1A because the two values from each cell line don't match up but the legends indicate they are similar experiments? E.g. surviving fraction at 2gy
- Fig 1A. How does one perform a paired T-test on two different cell lines? Is it done as before/after treatment on the same cells? Are they dependent samples or independent?
- Du145 were statistically different in terms of their growth by 4 days in S1C but then insignificant in Fig 1B by confluence? Is this qualitatively determined? Methods indicate two strategies of measuring 'confluence' but no details provided. Does the fluorescent labeling affect their growth dynamics? Seems to be the case.
- Fig 2a what are the statistics comparing, 2-Way ANOVA of each compared to parental?
- Fig 2a What is the purpose of showing images from day 12 and day 15 spheroids (PC3 and du145, respectively) when they are barely significantly different from each other based on the right hand graph? Do the authors intend to show significant differences in growth?
- Fig 4D and 4G are the legends wrong? The data are very confusing comparing it to the earlier studies in figure 2
- Fig 4A vs. 4B and 4E. What is the difference between 'spheroids controlled' and 'controlled spheroids' and is there another way to phrase this? What are controlled spheroids I can't find that in the text.
- Where is figure legend for figure 4 E,F,G?
- Based on figure 2B the parental cells of PC3 are 'outcompeted' by a margin of what appears to be ~3:1 (33% vs. 66%) at day 11. The images in S6 appear that, for the most part, parental cells outcompete the RR by almost the same margin. Were the colors misinterpreted or mislabeled?
- Not really appropriate to call the space between cancer cells 'stroma'. Stroma in the biological context contains connective tissue, and these cells all appear to be connected to each other in the absence of a true 'stroma' or extracellular matrix. Unless the cells are producing collagen or some other connective matrix?
- Methods indicates that cells were grown for up to 48 days after irradiation. Are there more details they can provide as to the culture conditions? The methods don't describe anything about how often the spheroids were fed, nor for other experiments, forget about irradiation. Methods needs significantly more detail with regards to culturing conditions.
- The notion of 'regrowth' seems entirely arbitrary and really confusing in this context. I would encourage the authors to choose some other method of analysis. The arbitrary definition of 2.5 fold in growth after irradiation as some metric of 'regrowth' is confusing and misleading. Shouldn't any amount of growth above the starting volume constitute 'regrowth' or 'growth'?

- From the methods section: "The experiment was repeated on a smaller scale using single doses of radiation on parental, mixed, and RR spheroids from both PC3(n = 37/ group; cut-off > 3.5) and DU1459(n = 20/ group; volume cut-off > 3.0) cell lines."
What's a 'smaller scale' and are the authors saying there are different 'cut-off' thresholds for what constitutes 'regrowth' depending on the experiment? Why not just define the growth rate? Or the slope of growth? Something unbiased would be more suitable in this context.
- Other studies have been published that contrast the evidence presented here. The authors should extrapolate their findings against those that have shown cooperation between mutant phenotypes can give rise to resistance (doi: 10.1371/journal.pcbi.1007278) or put them in the context of the microenvironment in which metastasis can be conferred between tumor and immune cells through cooperative or mutualistic behavior (doi: 10.1038/s41556-019-0346-x)

Reviewer #3:

Remarks to the Author:

In this paper the authors use experiments and mathematical modeling to study how interactions between heterogeneous populations could impact growth and response to radiotherapy in prostate cancer. The study is based on similar observations that were made in chemotherapy.

The main results are that reciprocal interactions between populations in mixed cultures enhance overall growth and reduce radiation sensitivity. These outcomes are hypothesized to originate from oxygen constraints and from cellular cross-talk that alter the tumor microenvironment. The findings suggest that ecological-type interactions are important in radiation response and could be targeted to reduce local recurrence.

The paper is very well written. The supplementary documents are comprehensive and provide a detailed description of the methodologies as well as some supporting data and figures.

Overall, I think that the paper is excellent and it merits publication in this journal. Several specific comments:

1) My main concern is with regards to the mathematical modeling. The mathematical model that the authors are basing much of their work is a system of two ordinary differential equations. It is a rather simple system in which two populations compete with each other. The choice of how to approach this competition is not very standard. Other works (say in the general context of competition between tumor cells and healthy cells) model competition differently, e.g., through the carrying capacity of the populations (arguing that having several populations occupying the same space does reduce the overall carrying capacity for each population, as long as both populations co-exist). Nevertheless, the current approach is still reasonable. The second part in which a hybrid model using a cellular automata in conjunction with a PDE is even less anchored in reality, but as a toy model it works. What I mean by "reality" is the deliberate omission of any detail. This leads, for example, to conclusions about the microenvironment and cell-to-cell communication without these mechanisms being actually modeled. So - yes - for the purpose of the present work it is ok to use such models but there should be additional discussion about their limitations. (How about discussing the nature of the model that should be written in order to obtain what you are aiming at - a control on the radiation plan that will be leading to the desired impact).

2) I don't understand the observation on lines 11-13 of page 5. Why in mixed DU145 spheroids, cell death was enhanced by 26% for the radioresistant population on day 5 resulting in an early growth advantage for the parental population?

3) On page 15 of the supplementary material - what do you mean by "best matched the images for both populations by visual inspection."? This comparison can (and should) be done using quantitative techniques.

4) minor point: on page 2, line 24, remove the "(i.e., α and β)". That doesn't mean anything to people that don't know what these parameters are.

Authors' responses to Reviewers' comments

We thank the reviewers for their highly positive and constructive comments on the study. We have modified our manuscript in line with the reviewers' suggestions. We detail below the comments that were raised, together with our responses (in blue text) and changes to the manuscript (in red text). We note that the cited page numbers refer to the manuscript with tracked changes.

Reviewer #1 (Remarks to the Author):

This is an outstanding investigation that blends seamlessly theoretical and empirical work in a way that allows the whole to be greater than the sum of its parts. It is an interesting and important subject. The manuscript is beautifully written. While I might quibble with some of the details and suggest some additional experiments, my overall recommendation is to publish it as is. My congratulations to the authors for excellent work.

Reply: We thank the reviewer for their positive comments on the study and manuscript.

Reviewer #2 (Remarks to the Author):

The study is generally fascinating and deploys multiple scientific disciplines to try and unravel how heterogeneity is implicated in resistance to radiation therapy in prostate cancer. At the same time, there are a lot of areas that need more explanation, significant under described methods, requires attention to detail, and other information that should be provided, buttressed or explained to provide better clarity. It's very challenging to read given the number of issues that came up and all the areas that cause confusion. In some areas I feel like the authors have created that confusion for themselves. I've enumerated a number of concerns here that are addressable but need attention. More specifically:

Reply: We thank the reviewer for their positive comments and also for identifying areas which require more explanation and/or detail. As explained below, we have modified our manuscript in accordance with the reviewer's suggestions.

Resistance concerns:

1• More information on the generation of these 'radioresistant' cell lines is needed. Were they generated in the lab as per reference 23 and 24? Or were they given to the lab from the originating authors? If it's the latter, what are the culture conditions required to maintain 'resistance', how often are they tested for 'resistance'? The resistance profiles are dissimilar between this paper and the reference, so some clarification would be useful.

Reply: We agree that clarification regarding the cell lines would be helpful. Although we worked primarily with the fluorescent cell lines in this study, we provide information on both the unlabelled cell lines and the fluorescently-labelled cell lines below.

Unlabelled cell lines. The unlabelled parental and radiation resistant (RR) cell line pairs (PC3 and DU145) were obtained from the originating authors. Our data on the unlabelled cell lines (presented in SI Fig 1) show comparable results to those reported in the original papers. For the PC3, we found an RPF of 1.18 (SI Fig 1); the original paper (doi: 10.1158/0008-5472.CAN-13-1657) reported an RPF of 1.19. For DU145, since the original paper did not report RPF, we compare SF2 values we obtained to interpolated SF2 values from the paper. We found SF2 values of 0.55 +/- 0.08 (SD) for DU145 parental and 0.62 +/- 0.08 (SD) for DU145 RR (SI Fig 1); the original paper (doi: 10.18632/oncotarget.13576) reported interpolated SF2 values (assuming a log axis) of 0.4-0.5 for DU145 parental and 0.5-0.6 for DU145 RR.

The original papers did not report using special culture conditions (e.g., extra courses of radiation) to maintain the resistance of the cells; therefore, we did not use special conditions either. The original PC3 RR cell population was assayed using a clonogenic assay to have resistance for up to 3 months (personal correspondence with Dr. Stanley Liu). The original paper on the DU145 cell populations reported that “radiation resistance was assayed by clonogenic survival, and ... [that] the cells maintained this phenotype in culture for at least 4 months”. To preserve the radiation resistance phenotype of the unlabelled cell lines, we expanded and froze many aliquots of the original vials. We maintained unlabelled cells in culture for no more than 10 passages (~6 weeks).

Fluorescently-labelled cell lines. In order to study ecological-type interactions between populations in our manuscript, we transduced the original cell lines with fluorescence and used flow cytometry sorting and antibiotic selection to isolate pools of fluorescent cells, as described in Methods. We subsequently measured the resistance profiles of fluorescent cells (Fig 1) using clonogenic and proliferation assays (discussed further in the reply to point 2).

We did not use special culture conditions to maintain radioresistance in the labelled cell lines. The labelled cells were discarded after 10 passages (~ 6 weeks) in culture. We verified that resistance was maintained during this time in culture (please see reply to point 2).

To account for these points, we have amended our manuscript as follows:

- Methods, page 16, lines 4-5, added underlined text:
“We obtained unlabelled parental (sensitive) and radiation resistant populations from two prostate cell lines, PC3 and DU145, from the Liu laboratory (University of Toronto, Canada).”
- Methods, page 16, lines 16-20, added text:
“Unlabelled cells were cultured for up to 10 passages (~6 weeks) for transduction; labelled cells were cultured for up to 10 passages (~ 6 weeks). No additional courses of radiation were used to maintain resistance. We measured clonogenic survival at 2 Gy (SF2) to verify that the labelled cells maintained the resistance phenotype for up to 12 passages (~8 weeks) in culture.”

2• Biological characterization of the ‘resistant’ cells should be provided such that they can be compared to other evidence in the literature for ‘radio resistance’.

Reply: In a review on the generation and characterization of radioresistant cell lines from McDermott and colleagues (<https://doi.org/10.3109/09553002.2014.873557>), the clonogenic cell survival assay is considered the gold standard experiment for characterization of radiation resistance in cells. The review recommends at least one additional line of evidence for characterization. Indeed, a variety of additional lines of evidence, such as increased proliferation after radiation and altered DNA repair capacity, have been used to characterize radiation resistant cells. However, the results of these additional lines of evidence have been more variable across different radioresistant cell lines and less consistent than results from the clonogenic survival assay. In our study, for both the unlabelled (SI Fig 1) and labelled (Fig 1) cells, we found enhanced clonogenic survival of the radioresistant cells in response to radiation. Thus, our data of the unlabelled and labelled cells are already aligned with current best practice for characterization of radiation sensitivity.

Excepting the data presented in the SI Fig 1 (characterizing unlabelled cells), all data presented in the manuscript derive from fluorescently-labelled cell lines. Given that the unlabelled cells have been extensively characterized in published work, and given that we used only the fluorescently-labelled cells to study ecological interactions, further characterization of the unlabelled cells is beyond the scope of this manuscript. Instead, we now focus on the characterization of the fluorescently-labelled cells that are used to study ecological-type interactions.

To strengthen the evidence of the radioresistance phenotype for the labelled cells, we performed additional experiments on the labelled cells to address the reviewer’s questions regarding: a) the time frame that the resistance lasts in culture (raised in point 1), and b) whether the cells are cross-therapy resistant (raised in point 5). To address the first question, we performed a clonogenic survival assay at 2 Gy using labelled cells from ~6-8 weeks in culture. Our results show that the labelled cells maintain resistance up to 8 weeks in culture:

To address the second question, we measured the response of the labelled cells to cisplatin, since increased resistance to DNA damaging agents can be another piece of evidence to support acquired radioresistance. Our results show that the RR populations from PC3 and DU145 are more resistant to cisplatin-induced death than their respective parentals (see new Fig 1 below). Together, our experiments provide strong evidence of the resistance phenotype of the RR labelled cells, and this biological characterization is consistent with that shown for other radioresistant cells in the literature.

We have amended the main text and supplementary text as follows:

- Results, page 4, lines 16-20, added text:
 “Radioresistant populations were also more resistant to death induced by the DNA damaging agent cisplatin (PC3, parental IC_{50} = 6.5 [95% CI 4.9 to 7.9], RR IC_{50} = 13.0 [95% CI, 10.9 to 15.1], $P < 0.0001$; DU145, parental IC_{50} = 1.8 [95% CI 1.6 to 2.0], RR IC_{50} = 6.6 [95% CI 5.9 to 7.3], $P < 0.0001$).”
- New Figure 1, page 29, amended Fig 1a, added Fig 1c:

- Figure 1 legend, page 30, added underlined text:
 “**Figure 1. Radiation resistant cell populations from two prostate cell lines have lower radiation sensitivity than parental counterparts.** (a) Surviving fraction at 2 Gy, as determined by clonogenic assay, for fluorescently-labelled parental and radioresistant (RR) populations from PC3 and DU145 cell lines. Surviving fraction values were normalized to 0 Gy. Bar indicates the

median; each data point represents one biological replicate (n = 6). ** $P = 0.005$ for PC3 and $P = 0.001$ for DU145, as determined by unpaired, one-tailed, t-test. **(b)** Growth curves of parental and RR populations grown as monolayers without and with radiation. Each data point represents one biological replicate (n = 4). * $P_{adj} < 0.05$, ** $P_{adj} < 0.01$, *** $P_{adj} < 0.001$, mixed-effects model with Sidak correction. **(c)** Percent cell viability (normalized to untreated wells) of parental and RR populations grown as monolayers in response to increasing concentrations of cisplatin (n=3-4 experiments). *** $P < 0.001$ for PC3 and DU145, as determined by extra sum-of-squares F-test."

- **Methods, page 17, lines 12-22, added paragraph:**

"Response to cisplatin

To check whether the RR cells had altered DNA damage response, we measured the cell viability of each population in response to a range of concentrations of cisplatin using a modified cytotoxicity assay³⁸. Briefly, single cells (2×10^3 cells in 100 μL /well) were seeded as monolayers in a 96-well plate and allowed to attach for 36 hours before treatment. Increasing concentrations of cisplatin (made in 100 μL /well) were added to each well resulting in final volume of 200 μL /well; cisplatin (Sigma) was prepared fresh for each treatment by dissolving the powder in 0.9% sterile-filtered saline to a final concentration of 3.3 mM. Treated cells were then cultured for 72 h in cisplatin before they were fixed with 10% formalin. Cell confluence was measured using Incucyte Base Software (Sartorius) and reformatted to a concentration-response curve by normalizing cell confluence values to the untreated well."

- **Methods, page 21, lines 18-20, added text:**

"For cisplatin cytotoxicity assays, IC_{50} values were calculated using a normalized response, variable slope, dose-response model (Prism, 8.0, GraphPad) and evaluated for statistical significance using extra sum-of-squares F-test."

3• The data showing 'resistance' is completely underwhelming and not convincing, particularly S1B and C, with regards to generated resistance. Perhaps they should be called radio-treated? And in fact, in Suppl. Fig 6 it says parental and radio recurrent. Is RR radiation resistant or are these supposed to be radiation recurrent? Authors may want to consider calling these radio recurrent throughout because it's been confusing to contend with the data that they are indeed truly radioresistant.

Reply: The additional experiments outlined in our reply to point 2 strengthen the contention that the labelled RR cell populations are more radiation resistant than their parental counterparts (Fig 1). Since these populations meet the criteria for radioresistance outlined in our reply to point 2, we have kept the naming convention to radioresistant in this manuscript and corrected the name in SI Fig 6. We have amended the text to clarify that the term radioresistant is used in a relative manner, i.e., the radioresistant population is resistant relative to their parental counterparts.

- **Results, page 4, lines 22-24, added text:**

“The term ‘radioresistant’ is henceforth used to describe cell populations with intrinsically lower radiation sensitivity than their parental counterparts.”

4• Why so many different ways of showing ‘resistance’ that tend to conflict with one another? For example, the clonogenic assays in Figure S1 stand in stark contrast to the data in figure 2a.

Reply: The clonogenic assays in the original Figure S1 were performed by seeding single cells in monolayers and measuring their clonogenic capacity after exposure to a range of radiation doses. Figure 2a, however, shows the growth of 2000 cells seeded as spheroids, with no radiation treatment. These two assays do not measure the same biological effect.

To clarify this point (and in partial response to point 8 below), we have amended the text as follows:

- Results, page 4, lines 8-12, added underlined text:
“... the unlabelled radioresistant (RR) populations had significantly higher 2D clonogenic survival (SI Fig 1B) ... In both fluorescently-labelled cell lines, the 2D clonogenic survival of the radioresistant populations...”

5• Are these cell lines cross therapy resistant? This would be useful to test since, in relapsed and radiation refractory clinical cases, patients are sometimes treated with systemic therapies.

Reply: We thank the reviewer for this suggestion. Please see our reply in point 2.

6• DU145 appear to be more proliferative without radiation (S1C), can the treatment with 6gy be normalized to their baseline growth in order to conclude on proliferation rates after treatment? Perhaps then they won’t be significantly different. This causes concern for the contention that they are ‘resistant’

Reply: To compare differences in growth baseline and post-radiation growth of the unlabelled DU145 RR cells (grown as monolayers), we plotted the data in the original SI Fig 1c on a log-linear scale:

A comparison of the slopes using a least squares regression analysis indicates that the four slopes are significantly different from one another:

Group	Slope	95% C.I.
Parental (no IR)	0.32	0.28 – 0.35
RR (no IR)	0.33	0.29 – 0.37
Parental (after IR)	0.18	0.16 – 0.20
RR (after IR)	0.20	0.18 – 0.22

However, a further comparison of specific groups (parental no IR vs RR no IR; parental after IR vs RR after IR) revealed no significant differences between the slopes. Based on this analysis, we believe the data are inconclusive. The experimental design used to generate the data in SI Fig 1c has limited statistical power to distinguish differences in growth: we did not measure the same well over time (due to the nature of the experimental setup, as described in reply to point 11) and we did not sample enough time points for the level of variance in the data. Thus, given the inconclusive nature of this experiment, we have removed this data from the figure.

- Results, page 4, lines 8-9, added underlined text and deleted text:
 “Here, after verifying that unlabelled radioresistant populations were morphologically different (SI Fig 1A) and had significantly higher 2D clonogenic survival (SI Fig 1B) and proliferation (SI Fig 1C) after radiation than parental populations...”
- Revised SI Fig 1, page 2, deleted SI Fig 1c and revised figure legend:

“**Supplementary Figure 1.** Radioresistant (RR) populations from unlabelled PC3 and DU145 cell lines are morphological distinct from parental populations, and have enhanced clonogenic survival and proliferation after radiation. **(a)** Morphology of parental and RR populations from PC3 and DU145 cell lines. **(b)** Clonogenic survival of PC3 (n = 4 experiments) and DU145 cells (n = 3 experiments) measured 10 days after irradiation with 0, 2, 4, and 6 Gy, and fitted to the linear-quadratic equation. RPF = radiation protection factor. Significance was evaluated using Student’s t-test (paired, one-tailed, $\alpha=0.05$) on area-under-the-curve values. **(c)** Growth curves of parental and RR populations from PC3 (n = 3 exp) and DU145 (n = 6 exp) cell lines.”

Other issues:

7• Figure 2c the data indicate that du145 RR cells die at a rate of 20-40% every 5 days in monotypic culture, but they tend to grow in other experiments. Why? Also, it’s perplexing that the monotypic and heterotypic cultures are insignificantly different in this panel even though the other data that are clearly less different are shown as statistically significant. How is that possible?

Reply: With regard to the first question, the net growth rate of a spheroid depends on the balance between the birth and death rates of the cells within the spheroid. During the initial phase of spheroid growth, growth is typically exponential because all cells receive a plentiful supply of nutrients (e.g., oxygen) and, as a result, are able to proliferate. As the spheroid continues to grow, nutrient levels towards the spheroid centre decrease, leading to reduced rates of cell proliferation, growth arrest, and, eventually, the emergence of necrotic or apoptotic cell death. During this period, the width of the outer rim of proliferating cells reduces due to nutrient limitations (e.g., oxygen diffusion), while the size of the secondary necrotic core increases as dead cells accumulate in the spheroid centre. The overall effect of these changes is a reduction in the net growth rate of the spheroid. When the net rate of cell death (at the spheroid centre) balances the net rate of cell proliferation, the spheroid attains its equilibrium volume at which the net growth rate is zero. Therefore, spheroids in conventional growth assays (such as the one used here) routinely comprise both proliferating cells and secondary necrotic cores. This effect is commonly reported in the literature and has been the subject of many mathematical and biological investigations (see references below). Thus, although the RR cells die at a rate of 20-40% every 5 days in homogeneous spheroids, they have a relatively constant proliferation rate of about 20-25% (Fig 3c) during this time. Inspection of the growth curve also shows that the RR spheroids only reach the inflection point (i.e., the point of peak growth) at day 15, which is consistent with the growth rate superseding the death rate during the time of the experiments.

Experimental References:

1. Folkman J. et al. Self-regulation of growth in three dimensions. *J Exp Med* (1973), 138: 745–753. doi: <https://doi.org/10.1084/jem.138.4.745>
2. Mueller-Klieser W. Multicellular spheroids. *J Cancer Res Clin Oncol* (1987), 113: 101–122. <https://doi.org/10.1007/BF00391431>
0. Friedrich J. et al. Experimental anti-tumor therapy in 3D: spheroids—old hat or new challenge? *Int J Radiation Biol* (2007), <https://doi.org/10.1080/09553000701727531>
1. Hirschhaeuser F. et al. Multicellular tumor spheroids: an underestimated tool is catching up again. *J. Biotechnol* (2010), doi: 10.1016/j.jbiotec.2010.01.012
3. Hari N. et al. Optical coherence tomography complements confocal microscopy for investigation of multicellular tumour spheroids. *Sci Rep* (2019), 9: 10601 <https://doi.org/10.1038/s41598-019-47000-2>

Modelling References:

1. Marušić M. et al. Analysis of growth of multicellular tumour spheroids by mathematical models. *Cell Proliferation* (1994), 27: 73-94. doi:10.1111/j.1365-2184.1994.tb01407.x
2. Casciari J.J. et al. Mathematical modelling of microenvironment and growth in EMT6/Ro multicellular tumour spheroids. *Cell Proliferation* (1992), 25: 1-22. [doi:10.1111/j.1365-2184.1992.tb01433.x](https://doi.org/10.1111/j.1365-2184.1992.tb01433.x)
3. Roose T. et al. Mathematical Models of Avascular Tumor Growth. *SIAM Review* (2007), 49:2, 179-208. <https://doi.org/10.1137/S0036144504446291>

4. Araujo R.P. et al. A history of the study of solid tumour growth: the contribution of mathematical modelling, *Bulletin of Mathematical Biology* (2004), 66(5): 1039-1091, <https://doi.org/10.1016/j.bulm.2003.11.002>
5. Greenspan H.P. Models for the Growth of a Solid Tumor by Diffusion, *Studies in Applied Mathematics* (1972), 51, <https://doi.org/10.1002/sapm1972514317>

With regard to the second question, in Fig 2c, the RR isolated from homogeneous vs mixed DU145 spheroids are significantly different at day 5 (using a binomial regression with FDR cut-off of 0.05). As described in our supplementary methods (Section 3, Statistical Analysis, page 7, Table 2), the statistical model was a poor fit for RR cells on day 15. Therefore, we did not calculate a *P*-value for this sample on this day, as the value would not be properly calibrated. We have now included this information in the figure legend for clarity (see below for changes). We are unsure what other data the reviewer refers to, but the growth differences between homogeneous and mixed cultures are statistically significant (please see response to point 13).

- Figure 2 legend, page 32, added text:

“No *P*-value was calculated for RR populations from DU145 spheroids on day 15 due to a poor statistical model fit.”

8• The flow of data is a bit confusing -- the growth dynamics in the absence of radiation pressure should be addressed first and then the mixed culture experiments with radiation would make more sense to bring in. Otherwise, it's like we're constantly going back and forth and confusing for the reader.

Reply: We thank the reviewer for this suggestion. The data are already presented in this way: Figures 2-3 assess growth dynamics in absence of radiation, Figures 4-5 present mixed culture experiments with radiation. However, we agree that the text does not make this flow of data very clear. To improve clarity, we have made the following changes:

- Results, page 5, line 3, added underlined text:

“Mixed cultures of tumour populations have enhanced growth and altered survival in the absence of radiation. To assess whether mixed cultures had altered growth characteristics, we measured the growth of unirradiated 3D tumour spheroids...”

- Results, page 5, line 23, added underlined text:

“Thus, despite the contrasting population dynamics between the two cell lines, these results indicate that the overall tumour growth rate, and survival of parental populations in particular, are enhanced in unirradiated tumours with mixed populations.”

- Results, page 6, line 1, added underlined text:

“Increased growth of unirradiated mixed tumours results from ecological-type interactions.”

- Results, page 7, line 17, added underlined text:

“Although the mechanism varies between the cell lines, these results support the premise that ecological-type interactions between heterogeneous tumour populations may enhance tumour growth in the absence of radiation.”

- Figure 2 legend, page 31, added underlined text:

“Figure 2. Inclusion of radioresistant cell populations enhances growth of unirradiated mixed 3D cultures and alters survival of each population...”

- Figure 3 legend, page 33, added underlined text:

“Figure 3. Mathematical modelling predicts the type of ecological interactions in unirradiated mixed prostate spheroids...”

- Figure 4 legend, page 36, added underlined text:

“Figure 4. Inclusion of radioresistant populations in spheroids reduces radiation sensitivity... (d,g) Percent of dead cells in each population isolated from irradiated homogeneous...”

- Figure 6 legend, page 39, added underlined text:

“Figure 6. Oxygen competition and cellular cross-talk mediate enhanced growth and reduced radiation sensitivity in unirradiated mixed PC3 spheroids... (e) Growth curves predicted solely by competition for space and oxygen in untreated homogeneous and mixed spheroids.”

9• What is the difference between S1B and Fig 1A because the two values from each cell line don't match up but the legends indicate they are similar experiments? E.g. surviving fraction at 2gy

Reply: As described above, SI Fig 1 characterizes unlabelled cells while Fig 1 characterizes labelled cells. The sensitivity of the labelled cells to radiation appears to be different, as assessed by a comparison of the SF2 values between the unlabelled cells and the labelled cells (see figure below). The change in radiation sensitivity by fluorescent labelling is not an issue in our study since we required populations to have differential sensitivity to study ecological-type interactions. However, our results highlight the impact fluorescent labelling can have on radiation sensitivity, and based on these analyses, we conclude that caution is needed when extrapolating results based on fluorescently labelled cells being cultured in vitro to unlabelled cells being studied in vivo.

Unlabelled PC3 (Welch's corrected, unpaired t-test): Fluorescently-labelled PC3 (unpaired t-test):

Unlabelled DU145 (unpaired t-test):

Fluorescently-labelled DU145 (unpaired t-test):

10• Fig 1A. How does one perform a paired T-test on two different cell lines? Is it done as before/after treatment on the same cells? Are they dependent samples or independent?

Reply: We thank the reviewer for noticing this error. We have now corrected it by performing an unpaired, t-test on the treated (i.e., radiated) samples comparing labelled RR vs parental. The figure and text have been modified, as highlighted in the reply to point 2.

11• Du145 were statistically different in terms of their growth by 4 days in S1C but then insignificant in Fig 1B by confluence? Is this qualitatively determined? Methods indicate two strategies of measuring 'confluence' but no details provided. Does the fluorescent labeling affect their growth dynamics? Seems to be the case.

Reply: The growth curves presented in the original SI Fig 1 vs Fig 1 were performed using different methods, which reflect the equipment available at the time the experiments were performed. In the original SI Fig 1, unlabelled cells were fixed at each time point, stained with Hoechst, and imaged using the Celigo cytometer; then, the number of nuclei was quantified as a measure of cell confluence using software included on the Celigo. In Fig 1, labelled cells were imaged over time using brightfield on the Incucyte Imaging System and analysed for confluence using Incucyte Base Software. Thus, both measurements are quantitatively determined. Difference in growth dynamics may have been due to fluorescent labelling, but may have also been due to the experimental setups used to measure growth.

We have removed the original SI Fig 1c for the reasons outlined in our reply to point 6, and have amended the Methods to provide clarity on the measurement of confluence for the labelled cells.

- Revised Methods, pages 16-17, line 23 onwards, added underlined text:
“Single cells (1×10^3 cells/well, 200 μ L medium) were seeded in flat-bottom 96-well plates, allowed to attach overnight, irradiated (0, 2, or 6 Gy), and fixed at various time points using 4% paraformaldehyde., and imaged daily using brightfield (Incucyte Live Cell Imaging System, Sartorius). Medium was changed every 2 days. Cell confluence or count was determined using Incucyte Base Software (Sartorius).”

12• Fig 2a what are the statistics comparing, 2-Way ANOVA of each compared to parental?

Reply: The statistics are comparing parental vs mixed, and parental vs RR.

- Figure 2 legend, page 31, added text:
“* $P_{adj} < 0.05$, ** $P_{adj} < 0.01$, *** $P_{adj} < 0.001$, 2-way ANOVA with post-hoc Bonferroni correction comparing parental vs mixed (blue asterisks) and parental vs RR (black asterisks) spheroids.”

13• Fig 2a What is the purpose of showing images from day 12 and day 15 spheroids (PC3 and du145, respectively) when they are barely significantly different from each other based on the right hand graph? Do the authors intend to show significant differences in growth?

Reply: Initially, we included the images to help the reader visually understand how the spheroids differ. However, we agree with the reviewer that these images do not add much value. Since 2D slices of spheres (such as spheroids) underestimate the differences in volumes, we have removed the images from the figure to avoid misrepresenting the spheroid volume data (see new Fig 2 below). For the same reason, we have also removed the images from Fig 4 (see new figure in reply to point 21).

The statistical values in the right hand graph show the comparisons of the spheroid volumes between parental vs mixed (blue asterisks), and of parental vs RR (black asterisks). These specific comparisons are statistically significantly different at the indicated time points. Importantly, the effect sizes among groups (see table below) show substantial differences. The changes listed in the replies to points 8 and 12 should further clarify the comparisons and the statistics.

Cell line	Day	Effect size-- parental vs mix (difference in volume)	Effect size--parental vs RR (difference in volume)
PC3	12	Mean parental: 0.55 +/- 0.12 mm ³ Mean mix: 1.16 +/- 0.18 mm ³ Mean difference: -0.61 mm ³ 95% C.I. of difference: -0.76 to -0.46 P < 0.0001	Mean parental: 0.55 +/- 0.12 mm ³ Mean res: 1.22 +/- 0.34 mm ³ Mean difference: -0.67 mm ³ 95% C.I. of difference: -0.95 to -0.40 P < 0.0001
PC3	15	Mean parental: 0.63 +/- 0.12 mm ³ Mean mix: 1.9 +/- 0.35 mm ³ Mean difference: -1.28 mm ³ 95% C.I. of difference: -1.55 to -1.0 P < 0.0001	Mean parental: 0.63 +/- 0.12 mm ³ Mean res: 1.67 +/- 0.40 mm ³ Mean difference: -1.04 mm ³ 95% C.I. of difference: -1.34 to -0.73 P < 0.0001
DU145	15	Mean parental: 0.59 +/- 0.04 mm ³ Mean mix: 0.76 +/- 0.19 mm ³ Mean difference: -0.17 mm ³ 95% C.I. of difference: -0.32 to -0.02 P = 0.03	Mean parental: 0.59 +/- 0.04 mm ³ Mean res: 0.72 +/- 0.10 mm ³ Mean difference: -0.13 mm ³ 95% C.I. of difference: -0.21 to -0.05 P = 0.002
DU145	17	Mean parental: 0.63 +/- 0.07 mm ³ Mean mix: 1.06 +/- 0.14 mm ³ Mean difference: -0.43 mm ³ 95% C.I. of difference: -0.55 to -0.32 P < 0.0001	Mean parental: 0.63 +/- 0.07 mm ³ Mean res: 0.89 +/- 0.08 mm ³ Mean difference: -0.26 mm ³ 95% C.I. of difference: -0.34 to -0.19 P < 0.0001

- Revised Fig 2 and legend (added underlined text and deleted text), pages 31-32:

“Figure 2. Inclusion of radioresistant cell populations enhances growth of untreated mixed 3D cultures and alters survival of each population. (a) Growth curves and representative images of 3D spheroids comprising parental, radioresistant (RR), or 1:1 mix of parental:RR tumour cell populations (n = 12 spheroids pooled from 2 experiments).”

14• Fig 4D and 4G are the legends wrong? The data are very confusing comparing it to the earlier studies in figure 2

Reply: We have amended the figure legends to clarify that Figure 2 shows data for untreated spheroids (in the absence of radiation pressure), while Figure 4 shows data for treated spheroids (after a dose of radiation). Please see response to point 8 for further details.

15• Fig 4A vs. 4B and 4E. What is the difference between ‘spheroids controlled’ and ‘controlled spheroids’ and is there another way to phrase this? What are controlled spheroids I can’t find that in the text.

Reply: There is no difference between the two phrasings. We used the term “spheroid controlled” to describe the number of spheroids that did not grow to a certain size, with the term “control” reflecting the curative outcome as measured *in vivo* in radiobiological/radiotherapeutic studies (namely, tumour control probability upon irradiation). Nevertheless, since the data are analysed in the same way as a Kaplan-Meier survival curve, we have now renamed this to “% spheroids below endpoint” and amended the axes of the relevant figures in Fig 4 (please see reply to point 21 for new figures).

16• Where is figure legend for figure4 E,F,G?

Reply: We thank the reviewer for noticing this. The letters e, f, and g were missing, and we have now corrected the legend (please see reply to point 21).

17• Based on figure 2B the parental cells of PC3 are ‘outcompeted’ by a margin of what appears to be ~3:1 (33% vs. 66%) at day 11. The images in S6 appear that, for the most part, parental cells outcompete the RR by almost the same margin. Were the colors misinterpreted or mislabeled?

Reply: The colours in SI Fig 6 are correct as shown: parental cells in green and RR cells in magenta (to stay consistent with the colour scheme in the rest of the manuscript). Since pseudo-colour in microscopy images can make it difficult to distinguish differences, we have now quantified the number of pixels from each cell population within each spheroid cross-section. This analysis shows that RR pixels comprise ~2 times more of the spheroid cross-section than parental pixels (see new SI Fig 6 below), supporting the flow cytometry results from Figure 2B. It is important to note that microscopy only presents a cross-section of the spheroid, while flow cytometry presents results from the whole spheroid.

We have made the following changes to address this point:

- Methods, page 20, lines 18-20, added text:
“To quantify the ratio of parental to radioresistant populations in spheroid cross-sections (n =16 spheroids), we measured the number of pixels from each population (i.e., signal) by applying a threshold value 5 times higher than the median value of the background (i.e., noise).”
- Methods, page 22, lines 2-3, added text:
“For quantification of population proportions in microscopy images, pixel numbers were analysed using a two-tailed, Wilcoxon matched-pairs signed rank test.”
- New SI Figure 6 and legend, SI page 7:

a

Parental/ RR

b

“**Supplementary Figure 6. (a)** Images showing spatial localisation of parental (green) and RR (magenta) cell populations in 16 different mixed PC3 spheroids isolated on day 11. Scale bar = 100 μ m. **(b)** Quantification of pixels from parental and RR in each spheroid cross-section ($n = 16$ spheroids). The four points shown in open circles are paired values from two spheroids that were imaged with a lower bit-resolution (8 instead of 16), resulting in lower pixel values. *** $P < 0.001$ by two-tailed, Wilcoxon matched-pairs signed rank test. Line indicates the median value.”

18• Not really appropriate to call the space between cancer cells ‘stroma’. Stroma in the biological context contains connective tissue, and these cells all appear to be connected to each other in the

absence of a true 'stroma' or extracellular matrix. Unless the cells are producing collagen or some other connective matrix?

Reply: We agree; indeed, there are no stromal cell compartments in these spheroids. However, it is known that tumour cells in spheroids can produce extracellular matrix. In the case of the present study, since 5% Matrigel was added to promote spheroid formation, the spheroids do contain some type of extracellular matrix. However, since we neither explicitly measured the amount of extracellular matrix nor the self-production of extracellular matrix by the tumour cells themselves, we have modified the text as suggested.

- Results, page 10, line 2, deleted text:
“Spheroids comprising mixed and radioresistant cells had reduced stromal staining and were visually less compact than those comprising parental cells (Fig 6A).”

19• Methods indicates that cells were grown for up to 48 days after irradiation. Are there more details they can provide as to the culture conditions? The methods don't describe anything about how often the spheroids were fed, nor for other experiments, forget about irradiation. Methods needs significantly more detail with regards to culturing conditions.

Reply: The culture conditions for the spheroids were described in Methods of the original text under “Cell culture”. However, we note that this information was not very clearly described. Since we used the same culture conditions for all spheroid experiments, we have now moved information regarding all spheroid culture to a separate section in the Methods and amended the text to improve clarity.

- Methods, pages 17-18, line 25 onwards, added text:
“*Spheroid generation and culture*
Homogeneous and mixed spheroids were generated in 96-well, ultra-low attachment plates (7007, Corning) by seeding different ratios of parental and radioresistant cell populations (2×10^3 total cells/well) using Matrigel (5% v/v, Corning) to promote spheroid formation³⁸. For all spheroid experiments, after an initiation phase of 3 days, spheroids were fed every 2 days by replacing 50% of the medium in each well with fresh medium (200 μ L total/well). Culture medium and incubation conditions were as described under “Cell culture”. Spheroid volumes were calculated using SpheroidSizer³⁹.”

20• The notion of 'regrowth' seems entirely arbitrary and really confusing in this context. I would encourage the authors to choose some other method of analysis. The arbitrary definition of 2.5 fold in growth after irradiation as some metric of 'regrowth' is confusing and misleading. Shouldn't any amount of growth above the starting volume constitute 'regrowth' or 'growth'?

Reply: Please see reply to point 21.

21• From the methods section: “The experiment was repeated on a smaller scale using single doses of radiation on parental, mixed, and RR spheroids from both PC3(n = 37/ group; cut-off > 3.5) and DU1459(n = 20/ group; volume cut-off > 3.0) cell lines.” What’s a ‘smaller scale’ and are the authors saying there are different ‘cut-off’ thresholds for what constitutes ‘regrowth’ depending on the experiment? Why not just define the growth rate? Or the slope of growth? Something unbiased would be more suitable in this context.

Reply: In response to the first question, the initial experiment (presented in Fig #) was performed with 7 radiation doses and 4 groups of spheroids (parental, 9:1 mix, 1:1, mix, RR). We then repeated the experiment using only 1 radiation dose and 3 groups of spheroids, which we called a “smaller scale”. We agree that this is not clear and have re-written the text (see below for changes).

In response to questions 2 and 3 : When a cell population is irradiated, the population generally experiences reduction in size and a delay before exponential growth occurs. We assume that the reviewer is referring to the rate of this exponential growth in question 2. When the size of a cell population is plotted (on a logarithmic scale) against time (on a linear scale), then in the phase of growth where the exponential growth dominates (e.g., days 10-18 in figure 3 from Demidenko, 2010, reproduced below), the size of the cell population appears linear with time. The slope of this line (beta in figure 3 from Demidenko, 2010) in log-linear space is the rate of growth of the exponential in linear space. We assume that the reviewer is referring to this slope, beta, in question 3, and therefore questions 2 and 3 are referring to the same quantity, which we will call growth rate here.

Figure 3 from Demidenko (2010):

It is possible for two cell populations to have the same growth rate, but experience different delays until exponential growth dominates. This phenomenon is well known in radiobiology and is described for example in Hall and Giaccia (2006) and Demidenko (2010). Growth rate as a metric is not sensitive to this delay; instead, the tumour growth delay reflects biological differences in radiation response that we wish to measure. Therefore, growth rate alone is insufficient and we must choose a different metric that

is both sensitive to differences in growth rates and differences in delays until the exponential phase of growth in a cell population dominates.

Such a metric is defined by Hall and Giaccia (2006) as “regrowth”: the time it takes for a tumour to grow to a specified volume (i.e. “cut-off”) above the initial volume following a course of radiation. As can be seen in figure 3 from Demidenko (2010), when the growth rate is not affected by radiation treatment, then the choice of cut-off does not affect the measured regrowth, as long as the cut-off is at a volume where exponential growth dominates for the cell population. Only when growth rate is affected by radiation, a departure from our null hypothesis, does the choice of cut-off matter. If increased radiation reduces growth rate and increases the time until exponential growth dominates, then smaller cut-offs are more conservative. In our case, the sampling resolution of our data decreased with time, so for each experiment, in order to maximise the sampling resolution of our data, we chose the lowest cut-off that was still within the exponential growth phase of all spheroids in the experiment and that avoided negative values of growth delay. As a guide, we chose values that are within the range commonly used in other radiobiology papers (doi: 10.15252/emmm.201809342; doi: 10.1158/1535-7163.MCT-17-0480; doi: 10.1158/0008-5472.CAN-13-1657)

In summary, our choices of cut-offs are: i) unbiased under the null-hypothesis and departures from the null-hypothesis where growth rate remains unchanged, ii) conservative when growth rate is reduced by increased radiation, iii) made using a simple rule that maximizes the power of our analyses.

References:

1. Hall, E. J., & Giaccia, A. J. (2006). *Radiobiology for the Radiologist* (Vol. 6).
2. Demidenko E. Three endpoints of in vivo tumour radiobiology and their statistical estimation. *Int J Radiat Biol.* 2010;86(2):164-173. doi:10.3109/09553000903419304

To make this concept more evident in our results and to improve consistency among figures, we have reanalysed the regrowth experiments from Fig 4 and plotted the data as survival curves to show radiation-induced growth delay in each spheroid group (i.e., parental, mix, RR). This was done by: 1) log-transforming the data, 2) estimating the average time it took for untreated spheroids to reach the volume endpoint (i.e., the cut-off), and 3) subtracting this averaged time from the time it took each irradiated spheroid to reach the volume endpoint. In doing this analysis, we made the following changes. First, we increased the threshold for DU145 from 3 to 4 to avoid having negative values of growth delay (some irradiated RR spheroids reached the threshold sooner than the average time taken for untreated RR spheroids). Second, we included regrowth data from DU145 irradiated at 6 Gy, in addition to at 10 Gy. During reanalysis, we noted that the growth curves of all groups from one plate of PC3 spheroids irradiated at 6 Gy did not resemble those from any of the other 9 plates. As this might be due to an experimental error, we removed this experimental batch from the analysis and modified the original figure (Fig 4b) accordingly.

We have made the following changes to the main text and figures to address these points:

- Results, pages 7-8, line 20 onwards, replaced deleted text with underlined text:

“To determine whether mixed cultures had altered radiotherapy response, we irradiated homogeneous and mixed spheroids at a range of doses and measured the time of growth to endpoint, as assessed by radiation-induced growth delay. their regrowth. Compared to PC3 spheroids comprising 100% parental cells [SCP₅₀, 5.17 Gy; 95% CI, 5.01 5.32 Gy], the dose required to prevent 50% of the spheroids from growing increased by 2 Gy with the inclusion of 10% radioresistant cells [SCP₅₀, 7.21 Gy; 95% CI, 7.06 7.36 Gy] and by another 2 Gy with the inclusion of 50% radioresistant cells [SCP₅₀, 9.56 Gy; 95% CI, 9.54 9.60 Gy] in mixed spheroids. Spheroids comprising 100% radioresistant cells had the highest dose required for control [SCP₅₀, 11.14 Gy; 95% CI, 11.14 11.15 Gy] (Fig 4A). The time of regrowth after radiation At each dose of radiation, growth delay curves differed significantly (on a global scale) among the four spheroid groups (parental, 9:1 parental:RR, 1:1 parental:RR, and RR). We subsequently tested whether inclusion of 10% radioresistant cells could reduce growth delay. Compared to PC3 spheroids comprising 100% parental cells, mixed spheroids (seeded 9:1 parental:RR) had significantly decreased growth delay after irradiation at 7.5 Gy ($P_{adj} = 0.005$; Fig 4A) and 10 Gy ($P_{adj} = 0.02$; Fig 4A). In a separate experiment using only 6 Gy radiation, the growth delay after radiation was reduced in mixed PC3 spheroids (seeded 1:1; $P_{adj} < 0.001$; Fig 4B), although radioresistant cells likely drove regrowth because they comprised $61 \pm 5.6\%$ and $79 \pm 5.6\%$ of cells in mixed spheroids on days 10 and 15, respectively (Fig 4C). While cell death of parental cells isolated from mixed spheroids on days 10 and 15 was $> 22\%$ lower ($P_{adj} = 0.045$) than that of those from homogeneous spheroids (Fig 4D), proliferation (of both current and cycled cells) reduced by 46% with an increase within G₂M/EdU⁻ ($P_{adj} < 0.001$), suggesting cell cycle arrest (SI Fig 4). Relative to parental cells isolated from homogenous spheroids, those isolated from mixed spheroids on days 10 and 15 had a $> 22\%$ reduction in cell death ($P_{adj} = 0.045$; Fig 4D). However, EdU⁺ uptake decreased by 46% and the proportion of cells in G₀G₁/EdU⁻ and G₂M/EdU⁻ phases increased ($P_{adj} < 0.001$; SI Fig 5), suggesting arrest in these cell cycle phases.

Similar results were obtained for mixed DU145 spheroids. The time of regrowth for mixed spheroids after 10 Gy radiation was reduced ($P_{adj} < 0.001$; Fig 4E) and After irradiation at 6 Gy and 10 Gy, the growth delay for mixed spheroids was reduced compared to parental spheroids (6 Gy, $P_{adj} < 0.01$; 10 Gy, $P_{adj} < 0.001$; Fig 4E). Regrowth at 10 Gy was not due to the radioresistant population dominating, as the parental population comprised $> 60\%$ of the mixed spheroids (Fig 4F). Furthermore, relative to the parental population in homogeneous spheroids, those in mixed spheroids had a 40% decrease in cell death in mixed spheroids on day 10 ($P_{adj} = 0.045$; Fig 4G), but a cell cycle arrest (as evidenced by a 19% reduced EdU⁺ uptake and an increase in G₂M/EdU⁻ phases ($P_{adj} = 0.045$; SI Fig 5)). In contrast, compared to the radioresistant population from homogeneous spheroids, those from mixed spheroids had a 37% increase in cell death on day 15 ($P_{adj} = 0.038$; Fig 4G), but no significant changes in EdU⁺ uptake or in G₂M/EdU⁻ proportions (SI Fig 5). Together, these data indicate that radioresistant populations not only reduces the growth delay upon irradiation the regrowth time after radiation, but also enhances the survival of the parental population in the irradiated mixed spheroids.”

- Methods, page 19, lines 5-22, added text:

“After log-transforming the volume data, we calculated the radiation-induced growth delay (in days) relative to untreated spheroids as the time for each irradiated spheroid to reach a volume endpoint (2.5 times the starting volume right before irradiation); we selected the lowest endpoint that was still within the exponential growth phase of all spheroids in the experiment. The average time for untreated spheroids to reach endpoint was estimated in R by local regression using the *loess* function with “direct” surface estimation to allow extrapolation for the parental spheroids (R project, v. 3.6.2).

Regrowth experiments were repeated by irradiating day 3 spheroids from three groups (parental, mixed, and RR) of PC3 cells (6 Gy, n = 17-18/ group) and of DU145 cells (6 Gy, n = 12/group; 10 Gy, n = 20/group). Spheroids were imaged using brightfield (Leica DM IRBE, Hamamatsu) for up to 27 days (PC3) and up to 23 days (DU145). The radiation-induced growth delay (delays) was calculated as above, but with different endpoints (3.5 times starting volume for PC3 and 4 times starting volume for DU145) to ensure the endpoint was within the exponential growth phase. Data from Fig 2 were used to estimate average time of untreated spheroids; the “span” parameter of the *loess* function was reduced from the default of 0.75 to 0.5 for the unirradiated DU145 spheroids to better estimate the average time of reaching the endpoint.”

- **Methods, deleted text:**

“For spheroid control probability experiments, dose-regrowth curves were fitted and analysed using a sigmoidal dose-response model (Prism 8.0, GraphPad). Values of spheroid control probability (SCP₅₀) are reported as the dose at which 50% of spheroids did not regrow, along with the 95% CI of the SCP₅₀.”

- **Methods, page 21, lines 20-24, added underlined text:**

“For post-radiation growth experiments, survival curves were analysed using the Mantel-Cox (log-rank) test and adjusted for multiple testing using Holm’s correction; spheroids that did not reach the endpoint during the timeframe of the experiment were marked as “censored” on the final day of experiment.”

- **New Fig 4 and Fig 4 legend, pages 35-36:**

“Figure 4. Inclusion of radioresistant populations in spheroids reduces radiation sensitivity. (a) Survival curves of time (days) taken for irradiated PC3 spheroids seeded with 100% parental (P) cells, 9:1 parental:radioresistant (RR), 1:1 P:RR or 100% RR cells to reach endpoint, adjusted for time taken by untreated spheroids to reach the same endpoint (n = 15 spheroids/group/radiation dose from 1 experiment). *** $P_{adj} < 0.001$, global log-rank test for all survival curves; + $P_{adj} < 0.05$, ++ $P_{adj} < 0.01$, log-rank test comparing survival curves of parental vs 9:1 P:RR (orange crosses). **(b, e)** Survival curves of time (days) taken for irradiated PC3 (6 Gy) and DU145 (6 and 10 Gy) spheroids seeded as parental, 1:1 P:RR, or RR to reach endpoint, adjusted for time taken by untreated spheroids to reach the same endpoint (n = 17 spheroids/group for PC3 pooled from 3 experiments; n = 12 spheroids/group for 6 Gy DU145 spheroids pooled from 2 experiments; n = 20 spheroids/group for 10 Gy DU145 spheroids from 1 experiment). *** $P_{adj} < 0.001$, global log-rank test for all survival curves; ++ $P_{adj} < 0.01$, +++ $P_{adj} < 0.001$, log-rank test comparing parental vs 1:1 P:RR (blue crosses) and parental vs RR (black crosses) spheroids. **(c, f)** Proportions of parental and RR populations measured over time using

flow cytometry from spheroids seeded as 1:1 mixture and irradiated (n = 4 experiments, $^{***}P_{adj} < 0.001$, 2-way ANOVA with Sidak correction). **(d, g)** Percent of dead cells in each population isolated from irradiated homogeneous spheroids or from mixed spheroids seeded as a 1:1 mixture of parental:RR (n = 4 experiments). $^{\dagger}FDR < 0.05$, overdispersed binomial regression with Benjamini-Hochberg correction. No *P*-value was calculated for parental populations from DU145 spheroids on day 15 due to a poor model fit.”

22• Other studies have been published that contrast the evidence presented here. The authors should extrapolate their findings against those that have shown cooperation between mutant phenotypes can give rise to resistance (doi: 10.1371/journal.pcbi.1007278) or put them in the context of the microenvironment in which metastasis can be conferred between tumor and immune cells through cooperative or mutualistic behavior (doi: 10.1038/s41556-019-0346-x)

Reply: In the second paragraph of the Discussion, we put our results into context with that of other studies investigating the impact of different ecological interactions (i.e., competition, mutualism, antagonism) in response to chemotherapy. Some of these studies report that competitive interactions can increase resistance in chemotherapy (doi: 10.1038/nature13556; doi:10.1038/s41467-017-01516-1), while others report that cooperative interactions can increase resistance in chemotherapy (doi: 10.1016/j.celrep.2014.06.045; doi: 10.1038/nature13187; doi: 10.1158/2159-8290.CD-14-1101). Although the *type* of interaction reported in our study is consistent with that reported in some but not all previous studies, all of these studies (including ours) show that ecological interactions (regardless of type) can increase resistance to cancer therapy. We limited our Discussion to the context of interactions between tumour cells, because interactions between tumour cells and stromal cells (whilst important) is beyond the scope of the current manuscript.

We modified the Discussion to clarify the differences in the findings and added in the first reference suggested by the Reviewer.

- Discussion, page 12, lines 16-19, added underlined text:
“Ecological interactions between tumour cell populations, irrespective of the type of interaction, appear to have evolutionary advantages for cancer. We found that both competitive and antagonistic interactions increase bulk tumour growth and resistance to radiation. Our results are consistent with findings from previous studies investigating the impact of ecological interactions in chemotherapy: although the *type* of interaction (e.g., competitive vs mutualism) can vary, ecological interactions appear to increase resistance to cancer therapy ^{14,17,27–29}. Furthermore, in our study, interactions increase the growth and post-radiation survival for the parental population in mixed spheroids in both cell lines.”

Reviewer #3 (Remarks to the Author):

In this paper the authors use experiments and mathematical modeling to study how interactions between heterogeneous populations could impact growth and response to radiotherapy in prostate cancer. The study is based on similar observations that were made in chemotherapy.

The main results are that reciprocal interactions between populations in mixed cultures enhance overall growth and reduce radiation sensitivity. These outcomes are hypothesized to originate from oxygen constraints and from cellular cross-talk that alter the tumor microenvironment. The findings suggest that ecological-type interactions are important in radiation response and could be targeted to reduce local recurrence.

The paper is very well written. The supplementary documents are comprehensive and provide a detailed description of the methodologies as well as some supporting data and figures.

Overall, I think that the paper is excellent and it merits publication in this journal. Several specific comments:

Reply: We thank the reviewer for the positive comments on the study and the manuscript.

1) My main concern is with regards to the mathematical modeling. The mathematical model that the authors are basing much of their work on is a system of two ordinary differential equations. It is a rather simple system in which two populations compete with each other. The choice of how to approach this competition is not very standard. Other works (say in the general context of competition between tumor cells and healthy cells) model competition differently, e.g., through the carrying capacity of the populations (arguing that having several populations occupying the same space does reduce the overall carrying capacity for each population, as long as both populations co-exist). Nevertheless, the current approach is still reasonable. The second part in which a hybrid model using a cellular automata in conjunction with a PDE is even less anchored in reality, but as a toy model it works. What I mean by "reality" is the deliberate omission of any detail. This leads, for example, to conclusions about the microenvironment and cell-to-cell communication without these mechanisms being actually modeled. So - yes - for the purpose of the present work it is ok to use such models but there should be additional discussion about their limitations. (How about discussing the nature of the model that should be written in order to obtain what you are aiming at - a control on the radiation plan that will be leading to the desired impact).

Reply: We thank the reviewer for their useful comments on our approach to mathematical modelling. Although we recognise that our models are simplistic, they are appropriate for the available data, allowing us to estimate values for the model parameters and, in so doing, to characterise the nature of the interactions between the different cell populations for the co-culture experiments. We note also that our Lotka-Volterra model is a natural generalisation of the modified logistic growth model proposed by the reviewer; the model can account for competitive, antagonistic and mutualistic interactions between the two cell populations (see Figure 3(b)). Indeed, our findings from the experimental data that interactions between the parental and RR DU145 populations are antagonistic was unexpected, and would not have been possible had we restricted attention to purely competitive models.

We used our hybrid CA model to show that although oxygen constraints can explain the spatial distribution of mixed spheroids they do not explain the growth dynamics (in terms of the growth curves). We then used biological experiments to demonstrate the existence of cell-cell communication between the PC3 cell populations.

The reviewer aptly noted that our spatial model does not account for cell-cell communication and is therefore unsuitable in its current form to test the hypothesis that microenvironmental pressures coupled with cell-cell communication can explain both spatial distribution and growth dynamics of the mixed spheroids. We intend to address this shortcoming in future work and in the meantime offer a short discussion on the possible improvements to the model, as per reviewer's suggestion.

- Discussion, pages 14-15, line 19 onwards, added text:
“While our spatially-resolved hybrid CA model was sufficient to study the impact of the microenvironment (i.e., oxygen constraints) on tumour development in a 2D cross-section of a tumour spheroid, the model in its current form also does not account for direct interactions between cell populations. To generate more realistic predictions, the model could be extended to three dimensional space, along with rules governing cell behaviour that are modified to account for interactions between different cell populations. Since cells can interact directly and indirectly via secretion of molecules, our model could be further modified to explicitly track factors released by coexisting cell populations. These modifications, whilst adding complexity to the model, would allow us to test various hypotheses pertaining to the impact of ITH on tumour growth and dynamics.”

2) I don't understand the observation on lines 11-13 of page 5. Why in mixed DU145 spheroids, cell death was enhanced by 26% for the radioresistant population on day 5 resulting in an early growth advantage for the parental population?

Reply: This was perhaps a speculative interpretation of the data. If the death of DU145 RR population was enhanced by 26% on day 5, then the end result would be an advantage for the parental population in terms of survival. We have modified the wording to reduce the speculation.

- Results, page 5, line 20, deleted text and added underlined text:
“However, in mixed DU145 spheroids, cell death was enhanced by 26% for the radioresistant population on day 5 ($P_{adj} = 0.01$, Fig 2C), which could have led to resulting in an early growth survival advantage for the parental population.”

3) On page 15 of the supplementary material - what do you mean by "best matched the images for both populations by visual inspection."? This comparison can (and should) be done using quantitative techniques.

Reply: We agree that, ideally, the comparison should be undertaken using quantitative techniques (e.g. least squares minimisation or Bayesian inference) to estimate parameters from the ABM by fitting its simulated data to the experimental data. In practice, this is computationally challenging when searching over a high-dimensional parameter space since, for every parameter choice, multiple realisations of the ABM must be generated (Lambert et al, 2018). For these reasons, and since the differences between the spatial distributions for the shortlisted models were marked, visual inspection was deemed sufficient to select the models. We postpone for future work a systematic analysis using Bayesian inference to fit the ABM to the experimental data.

References:

B Lambert, AL MacLean, AG Fletcher, AN Combes, MH Little, HM Byrne (2018). Bayesian inference of agent-based models: a tool for studying kidney branching morphogenesis. *Journal of mathematical biology* 76 (7), 1673-1697

4) minor point: on page 2, line 24, remove the "(i.e., α and β)". That doesn't mean anything to people that don't know what these parameters are.

Reply: Since we use these parameters in our study, we amended the text to clarify the meaning of these parameters.

- Introduction, page 2, line 25, added underlined text:
"Firstly, the linear-quadratic model used to predict tumour control probability contains single, fixed values for the radiation sensitivity parameters (i.e., α and β , the constants for cell killing in the linear and quadratic phases, respectively)."

Reviewers' Comments:

Reviewer #2:

Remarks to the Author:

I'm satisfied with the revisions. Thank you for taking the time to address each one individually and provide new experiments where needed. It is indeed much improved. Best of luck to all authors.

Reviewer #3:

Remarks to the Author:

My concerns have been addressed. I have no further comments.